# Evaluation of gas-particle partitioning in a regional air quality model for organic pollutants

C. I. Efstathiou[1], J. Matejovičová[1,2], J. Bieser[3,4], and G. Lammel[1,5]

[1]Masaryk University, Research Centre for Toxic Compounds in the Environment, Kamenice 5, 62500 Brno, Czech Republic
[2]Slovak Hydrometeorological Institute, Jeséniova 17, 83315 Bratislava, Slovak Republic
[3]Institute of Coastal Research, Helmholtz-Zentrum Geesthacht, Max-Planck-Str. 1, 21502 Geesthacht, Germany
[4]German Aerospace Center (DLR), Oberpfaffenhofen, 82234 Weßling, Germany
[5]Max Planck Institute for Chemistry, Multiphase Chemistry Department, Hahn-Meitner-Weg 1, 55128 Mainz, Germany

*Correspondence to*: C. I. Efstathiou (efstathiou@recetox.muni.cz)

**Abstract.** Persistent organic pollutants (POPs) are of considerable concern due to their well-recognised toxicity and their potential to bioaccumulate and engage in long-range transport. These compounds are semi-volatile and therefore partition between vapour and condensed phases in the atmosphere, while both phases can undergo chemical reactions. This work describes the extension of the Community Multi-scale Air Quality (CMAQ) modelling system to POPs with a focus on establishing an adaptable framework that accounts for gaseous chemistry, heterogeneous reactions, and gas-particle partitioning (GPP). The effect of GPP is assessed by implementing a set of independent parameterisations within the CMAQ aerosol module, including the Junge-Pankow (JP) adsorption model, the Harner-Bidleman (HB) organic matter (OM) absorption model, and the dual Dachs-Eisenreich (DE) black carbon (BC) adsorption and OM absorption model. Use of these descriptors in a modified version of CMAQ for benzo[a]pyrene (BaP), results in different fate and transport patterns as demonstrated by regional scale simulations performed for a European domain during 2006. The dual DE model predicted 24.1% higher average domain concentrations compared to the HB model, which was in turn predicting 119.2% higher levels compared to the baseline JP model. Evaluation against measurements from the European Monitoring and Evaluation Programme (EMEP) reveal the capability of the more extensive DE model to better capture the ambient levels and seasonal behaviour of BaP. It is found that the heterogeneous reaction of BaP with $O_3$ may decrease its atmospheric lifetime by 25.2% (domain and annual average) and near-ground concentrations by 18.8%. Marginally better model performance was found for one of the six EMEP stations (Košetice) when heterogeneous BaP reactivity was included. Further analysis shows that for the rest of the EMEP locations the model continues to underestimate BaP levels, an observation that can be attributed to low emission estimates for such remote areas. These findings suggest that, when modelling the fate and transport of organic pollutants on large spatiotemporal scales, the selection and parameterisation of GPP can be as important as degradation (reactivity).

# 1 Introduction

Polycyclic aromatic hydrocarbons (PAHs) are a group of lipophilic organic compounds with demonstrated carcinogenicity and potential to bioaccumulate (Diamond and Hodge, 2007; Finlayson-Pitts and Pitts, 1999; Pedersen et al., 2004; 2005; WHO, 2003). Typically emitted to the atmosphere as a mixture of semi- to low-volatile congeners with variable carcinogenic

potency, their fate and transport in the environment is difficult to assess. Current limitations involve inadequate knowledge of photochemistry (Keyte et al., 2013), air-surface exchange (Galarneau et al., 2014; Keyte et al., 2013; Lammel et al., 2009), and combustion sources (Bieser et al., 2012; Lammel et al., 2013). In addition, aluminium and steel production along with petrogenic sources are significant in some environments (Wenborn et al., 1999). This relationship to anthropogenic activity and population density raised awareness on their potential as health stressors, with hot spots identified in urban

industrial settings around the world (Šrám et al., 2013; Zhu et al., 2015). On the other hand, observations of PAHs at remote sites and global circulation model studies indicate long-range transport (Keyte et al., 2013; Sehili and Lammel, 2007). Therefore, and due to resistance to degradation, PAHs are classified as persistent organic pollutants (POPs) by the United Nations Economic Commission for Europe (UNECE) Convention on Long-range Transboundary Air Pollution (CLRTAP) and are listed in the Protection of the Marine Environment of the North-East Atlantic (OSPAR).

Much of the scientific literature is focussed on the 16 U.S. EPA priority PAHs i.e., acenaphthene (Ace), acenaphthylene (Acy), fluorene (Fln), naphthalene (Nap), anthracene (Ant), fluoranthene (Flt), phenanthrene (Phe), benzo[a]anthracene (BaA), benzo[b]fluoranthene (BbF), benzo[k]fluoranthene (BkF), chrysene (Chr), pyrene (Pyr), benzo[ ghi ]perylene (BgP), benzo[a]pyrene (BaP), dibenzo[a,h]anthracene (DBahA), and indeno[1,2,3-cd]-pyrene (IPy), albeit not well justified (Andersson and Achten, 2015). Used as a biomarker for the carcinogenic risk of PAHs, BaP is a criteria pollutant in many

countries, sometimes together with a few other bioaccumulative members. While in North America no federal guidelines are currently in effect, the European Union indicative limit value is set at 1 ng m$^{-3}$ of BaP (EC, 2004), and certain countries have established stricter air quality standards (i.e. 0.25 ng m$^{-3}$ of BaP in the United Kingdom) (DETR, 2009).

A number of studies, which aimed at exploring the atmospheric fate and transport of POPs on various scales, have been presented in the last two decades. These include relatively simple mass balance models developed under the Lagrangian

framework, some of which have been extended to multi-media applications for studying the accumulation and exchange of POPs between various compartments (Halsall et al., 2001; Lang et al., 2008; 2007; Liu et al., 2007; Prevedouros et al., 2004; 2008; Van Jaarsveld et al., 1997). Eulerian grid models add yet another level of complexity regarding process detail that is accompanied by additional computational demands. Global scale chemical transport models (Friedman and Selin, 2012; Lammel et al., 2009; Sehili and Lammel, 2007), as well as hemispheric models (Gusev et al., 2005; Hansen et al., 2006;

Shatalov et al., 2005) have been applied to study the long-range transport of POPs. Moreover, coupled meteorological and air quality modelling frameworks have presented efforts to further investigate the treatment of physical and chemical atmospheric processes related to POPs in higher resolution (Aulinger et al., 2007; Bieser et al., 2012; Cooter and Hutzell, 2002; Galarneau et al., 2014; Inomata et al., 2012; Matthias et al., 2009; Meng et al., 2007; San José et al., 2013; Silibello et

al., 2012; Zhang et al., 2009). Table S1 provides a summary of the most relevant processes covered in the respective regional model implementations and subsequent evolution tracked by author. While such studies represent the initial efforts to accurately model POPs on regional scale, important processes such as gas-particle partitioning, heterogeneous reactivity, and a multi-compartment approach for the relevant compounds are not always studied. The Weather Research Forecast along with the Community Multi-scale Air Quality for BaP (WRF-CMAQ-BaP) model presented in this article is considered to be the first effort to examine the full spectrum of gas-particle partitioning models relevant to POPs in the same regional-scale modelling application. The model attempts to capture POPs' aerosol chemistry and dynamics within the CMAQ framework and can be extended to account for mass exchange between adjacent compartments for compounds that such processes are deemed critical (i.e soil volatilization).

## 2 Model development

The model development framework is based on the CMAQ version 4.7.1 modelling system (Byun and Schere, 2006; Byun and Ching, 1999) with the CB05 gas phase chemistry mechanism (Yarwood et al., 2005) and the AERO4 version of aerosol module (Binkowski, 2003). In addition, CMAQ contains modules representing advection, eddy diffusion, in-cloud and below-cloud scavenging with precipitation. Advection and diffusion satisfy mass conservation and include removal by dry deposition, while in-cloud and precipitation processes simulate aqueous chemistry and wet deposition by cloud droplets (Byun, 1999a; 1999b). The aerosol module (Binkowski, 2003) determines concentrations of trimodal size distributed particulate material of diameter less than 10 μm. According to this model, each mode can be described as an internal mixture of several compounds; mixing state on the single particle level is not addressed, but is considered equivalent to internal mixing. Aitken and accumulation modes include particles of diameter ~2.5 μm or less (PM2.5), which are either emitted or produced by gas-to-particle conversion processes. A thermodynamic mechanism, based on temperature and relative humidity, determines air concentrations of water, sulfate, ammonium, and nitrate aerosols in the fine modes (Nenes et al., 1998; Zhang et al., 2000). Organic and elemental carbon species are also included in these modes. The third mode represents coarse material having diameters between 2.5 and 10 μm and includes dry inorganic material emitted by natural and anthropogenic sources. Other processes like coagulation, particle growth, aerosol-cloud interactions, and new particle formation are included in the treatment.

CMAQ adaptations are required to fully address the behaviour of semi-volatile organic compounds (SOCs) in the atmosphere and include additional homogeneous gas phase reactions, a modular algorithm to describe the mass exchange of SOCs between the gas and particulate phases, and a description of deposition for each phase state of the compound of interest. For the case of benzo(a)pyrene (BaP), one gas-phase species and three particulate-sorbed species, one for each aerosol mode, are introduced. It is assumed that only the gaseous molecule undergoes chemical degradation through a reaction with the hydroxyl (OH) radical, leading to an unspecified product that does not undergo further chemical reaction. A summary of the physico-chemical data used in this study is presented in Table 1. Furthermore, a module that accounts for

heterogeneous chemistry was included for compounds that such reactions represent a significant loss according to experimental studies (e.g. BaP reaction with $O_3$).

## 2.1 Modular implementation of gas-particle partitioning (GPP) in CMAQ

The distribution of POPs between the gas and the particulate phases is widely recognised as the most important parameter in
describing their fate and transport in the atmosphere. PAHs like benzo(a)pyrene can be adsorbed to mineral surfaces or elemental carbon, absorbed by organic aerosol, or absorbed by aerosol water (Lohmann and Lammel, 2004). Therefore, the aerosol module of CMAQ was enhanced with an array of models that assume different processes as determining gas-particle partitioning, i.e. an adsorption model (Junge-Pankow (Junge, 1977; Pankow, 1987)), an organic matter (OM) absorption model ($K_{oa}$ (Harner and Bidleman, 1998)), and a dual OM absorption and elemental or black carbon (EC/BC) adsorption
model (Dachs and Eisenreich, 2000; Lohmann and Lammel, 2004). In addition, a method for calculating the absorption in particulate phase water was included in the module, employed for all aerosol modes in a manner similar to the GPP processes (Aulinger et al., 2007; Cooter and Hutzell, 2002).

The new GPP algorithm for CMAQ brings together existing implementations for pesticides (Cooter and Hutzell, 2002), PAHs (Aulinger et al., 2007; San José et al., 2013), polychlorinated biphenyls (PCBs) (Meng et al., 2007), dibenzo-*p*-dioxins
and dibenzofurans (PCDD/Fs) (Zhang et al., 2009) in a unified module that allows for flexible GPP model combinations, making it applicable to a wide range of SOCs. The following assumptions were taken into consideration: instantaneous relaxation to phase equilibrium (no mass transport kinetic limitations), the compound does not irreversibly transform in the particulate phase, and sorption processes do not interact. By extending the gas-phase and modal aerosol treatments in CMAQ for SOCs, these assumptions allow the use of simple ratios to determine the concentration of each form following

$$a_i = \phi_i(a_i + g) \tag{1}$$

where $a_i$ is the particulate concentration in each mode (i=1=Aitken, 2=accumulation, 3=coarse), g is the gaseous concentration and $\phi_i$ is the particulate fraction of the compound in mode, with a general formulation (Aulinger et al., 2007; Cooter and Hutzell, 2002)

$$\phi_i = \phi_i^{ad} + \phi_i^{ab} + \phi_i^{aq} \tag{2}$$

This way, all partitioning processes: adsorption ($\phi_i^{ad}$), absorption in OM ($\phi_i^{ab}$), and absorption (solution) in aerosol water ($\phi_i^{aq}$) are simultaneously treated. Following the rule of mass conservation for the compound in both phases, the total
concentration $c_{tot}$ can be introduced

$$g = c_{tot} - a_1 - a_2 - a_3 \tag{3}$$

leading to a system of three linear equations

$$\begin{bmatrix} 1 & \phi_1 & \phi_1 \\ \phi_2 & 1 & \phi_2 \\ \phi_3 & \phi_3 & 1 \end{bmatrix} \cdot \begin{bmatrix} a_1 \\ a_2 \\ a_3 \end{bmatrix} = c_{tot} \times \begin{bmatrix} \phi_1 \\ \phi_2 \\ \phi_3 \end{bmatrix} \tag{4}$$

that yields the particulate concentrations $a_i$ in each mode. Subsequently, using the previous equation, the gaseous concentration g can be calculated.

### 2.1.1 Junge – Pankow adsorption model

The first GPP scheme is based on the model of exchangeable SOC adsorption to aerosols presented by Junge (1977) and later critically reviewed by Pankow (1987)

$$\phi_i^{ad} = c_J S_i / (p_L^\circ + c_J S_i) \tag{5}$$

The equation above relates the mass fraction of chemical adsorbed to particles in each mode i ($\phi_i^{ad}$) to the subcooled liquid vapour pressure of the compound ($p_L^\circ$, Pa) and the aerosol surface density ($S_i$, $m^2$ of particles' surface per $m^3$ of air). The parameter $c_J$ (unit, Pa m) depends on the chemical nature of both the adsorbant and the adsorbate, and cannot be expected to be a constant, as confirmed by field experiments (Lammel et al., 2010). Nevertheless, a value of $c_J$ = 0.172 Pa m has been suggested (Pankow, 1987) and is often used in modelling applications. The surface area parameter, $S_i$, which is a characteristic of the aerosol required for the Junge-Pankow model, is calculated at each time step for each mode within the aerosol module of CMAQ (Binkowski, 2003).

The empiric JP model, despite assuming adsorption to be the only relevant partitioning process, may eventually not be free from absorptive contributions (to $c_J$), but these should be very low. The fitting in the empiric parameterisation was done for all possible vapours, many of these not soluble in octanol. No $c_J$ for BAP derived from ambient measurements has ever been published. However, we found recently in background aerosol of Central Europe (Košetice 2012-13, n=162 aerosol samples, $c_J$ derived from experimental *S/V* data; Shahpoury et al., 2016) that $c_J$ for BAP is very low, namely 0.01-5.46 Pa cm (median 0.44 Pa cm), supporting the perception that a possible implicit absorptive contribution to $c_J$ must be very low.

### 2.1.2 $K_{oa}$ absorption model

The second GPP scheme follows the work of Pankow (1994; 2007), with refinements from Harner and Bidleman (1998) that described how the particulate fraction of a compound could be derived from the aerosol-air partitioning coefficient Kp. Following this approach for each mode i, the predicted particulate fraction $\phi_i^{ab}$ is linked to the total mass concentration of suspended particles $c_{TSP_i}$ with the following equation

$$\phi_i^{ab} = K_{p,i}^{OM} c_{TSP_i} / (K_{p,i}^{OM} TSP_i + 1) \tag{6}$$

Pankow (1987; 1998) demonstrated that the gas-particle partitioning coefficient, Kp, and the use of $p_L^\circ$ as its descriptor are valid both for adsorption and absorption of SOCs. In the case of absorptive partitioning, Pankow (1998) derived

$$K_{p,i}^{OM} (m^3 \mu g^{-1}) = \frac{f_{OM_i} 760RT}{10^6 M_{OM} \gamma_{OM} p_L^\circ} \tag{7}$$

where $f_{OM_i}$ is the mass fraction of OM in the particle that can absorb gaseous SOCs, R is the ideal gas constant (= 8.314 Pa $m^3$ $mol^{-1}K^{-1}$), T (K) is absolute temperature, $M_{OM}$ (g $mol^{-1}$) is the mean molecular weight of the OM phase, $\gamma_{OM}$ is the

activity coefficient of the selected SOC in the OM on a mole fraction basis, and $p_L^{\circ}$ (Pa) is the vapour pressure of the pure SOC (subcooled liquid in the case of solids). Partitioning coefficients normalized to the OM content were introduced to further support the absorption hypothesis (Finizio et al., 1997). Pankow (2007), and Harner and Bidleman (1998) replaced $p_L^{\circ}$ by the octanol-air partitioning coefficient, $K_{oa}$

$$K_{p,i}^{OM} = 10^{-12} \frac{f_{OM_i} M_{oct} \gamma_{oct} K_{oa}}{M_{OM} \gamma_{OM} \rho_{oct}} \tag{8}$$

where $M_{oct}$, $M_{OM}$ (g mol$^{-1}$) are the molecular weights of octanol (= 130 g mol$^{-1}$) and the OM phase, $\gamma_{oct}$, $\gamma_{OM}$ are the activity coefficients of the SOC in octanol and OM, respectively, and $\rho_{oct}$ is the density of octanol (0.82 kg L$^{-1}$). Assuming that octanol imitates organic matter in PM, Harner and Bidleman (1998) suggested that the ratio of $\gamma_{oct}/\gamma_{OM}$ and $M_{oct}/M_{OM}$ can be assumed to be 1. However, it was later suggested that $M_{OM}$ could be much higher, particularly in secondary organic aerosols containing polymeric structures (Kalberer et al., 2004); a mean value of 500 g mol-1 was later suggested by Götz et

al. (2007), which results in $M_{oct}/M_{OM}$ of 0.26. Under the assumptions that $\gamma_{oct}/\gamma_{OM}$ and $MW_{oct}/MW_{OM} = 1$, the equation above can be simplified to the following

$$log\, K_{p,i}^{OM} = log\, K_{oa} + log\, f_{OM} - 11.91 \tag{9}$$

### 2.1.3 Dual OM absorption and black carbon adsorption model

Finally, the third GPP scheme extends the previous OM absorption model to include adsorption onto particulate soot, according to evidence showing the measured $K_p$ to be exceeding the predicted $K_p$ based solely on absorption in OM (Dachs

and Eisenreich, 2000; Fernandez et al., 2002; Lohmann and Lammel, 2004; Ngabe and Poissant, 2003). Considering the contribution of those two additive processes a modified partition coefficient can be employed

$$K_{p,i}^{dual} = 10^{-12} \left( \frac{f_{OM_i} M_{oct} \gamma_{oct} K_{oa}}{M_{OM} \gamma_{OM} \rho_{oct}} + \frac{f_{BC_i} a_{atm-BC} K_{SA}}{a_{soot} \rho_{BC}} \right) \tag{10}$$

where $f_{BC_i}$ is the fraction of black carbon in the particulate matter, $a_{atm-BC}$ and $a_{soot}$ are the available surfaces of atmospheric black carbon (BC) and diesel soot, respectively, $K_{SA}$ is the partitioning coefficient between diesel soot and air. A density value of 1 kg L$^{-1}$ has often been assumed for BC, however values in the range of 1.7-1.9 kg L$^{-1}$ have been

measured and recommended by Bond and Bergstrom (2006). Thus, the equation can be simplified

$$K_{p,i}^{dual} = 10^{-12} \left( 0.32 \times f_{OM_i} K_{oa} + 0.55 \times f_{BC_i} K_{SA} \right) \tag{11}$$

This method is subject to uncertainties (Goss, 2004), but is accepted and suitable (Dachs et al., 2004; besides others). Soot-air partition coefficients ($K_{SA}$) were calculated as the ratio of soot-water adsorption constants $K_{SW}$ and the inverse Henry's Law constant (H′), with $K_{SW}$ values adopted from Bärring et al. (2002). As a step to address uncertainties contributed by different methods in calculating $K_{SA}$, a thermodynamic estimation model suggested by van Noort (2003) was also tested:

$$log K_{SA} = -0.85\, log\, p_L^{\circ} + 8.94 - log(998/a_{soot}) \tag{12}$$

The soot specific surface area was set to 18.21 m$^2$ g$^{-1}$, derived as the geometric mean of surface areas for traffic, wood, coal, and diesel soot (i.e. 59.4, 3.6, 8.2, and 62.7 m$^2$ g$^{-1}$, respectively) (Jonker and Koelmans, 2002). Additionally, in order to estimate uncertainties arising from K$_{oa}$ and its temperature dependence, two different parameterisations were tested, the first based on the work of Beyer et al. (Beyer et al., 2000) and the second following the temperature-dependent parameterisation

suggested by Odabasi et al. (2006).

## 2.2 Dissolution into aerosol water

An assumption of the CMAQ aerosol module is that organics influence neither the water content nor the ionic strength of the system. While this assumption may not be valid for many atmospheric aerosols, sufficient basic data are not available to treat the system in a more complete way. In a similar manner, the dissolution of PAHs in the aqueous electrolyte was calculated

using the inverse Henry's Law constant (Aulinger et al., 2007; Cooter and Hutzell, 2002; Lohmann and Lammel, 2004). Lohmann and Lammel (2004) derived the partitioning coefficient K$_{p,i}^{aq}$, although they conclude that as long as a solid-air interface exists, the contribution of dissolved PAHs to the gas-particle partitioning will always be negligible. To account for this, a "wet aerosol" switch was introduced and associated with the relevant partitioning coefficient

$$log\,K_{p,i}^{aq} = \; log\,K_{wa} + \; log\,f_{wa,i} - 12.0 \tag{13}$$

where f$_{wa,i}$ is the mass fraction of aerosol water in each mode i and K$_{wa}$ is the water-air partitioning coefficient, which is

equal to H′. The aerosol water content is computed in CMAQ based on a variation of the ZSR method (Byun and Ching, 1999; Kim et al., 1993). As suggested by Aulinger et al. (2007) for CMAQ, only if the ratio of ammonium sulfate to aerosol water drops below the solubility of ammonium sulfate, the aerosol is treated as wet and absorption into aerosol water takes effect, while adsorption to inorganic material is switched off.

## 2.3 Gas-phase and heterogeneous reactions

Reactions with the hydroxyl radical (OH), ozone (O$_3$), and the nitrate radical (NO$_3$) may determine the atmospheric fate and long-range transport potential of SOCs. Gas-phase reactions with OH are believed to be the dominant loss pathway for semi-volatile PAHs (3-4 rings), like anthracene and pyrene (Atkinson and Arey, 1994; Finlayson-Pitts and Pitts, 1999; Keyte et al., 2013). On the other hand, in the case of non-volatile PAHs (5 and more rings, like BaP), loss due to heterogeneous reaction with ozone is believed to dominate (Finlayson-Pitts and Pitts, 1999). Experimental studies of the kinetics of PAHs

sorbed to particles indicate strong influences of mixing state (single particle anisotropy), morphology, and phase state of the particles (Kwamena et al., 2004; Perraudin et al., 2007; Pöschl et al., 2001; Shiraiwa et al., 2009; Zhou et al., 2012). Yet, detailed heterogeneous chemistry models would require corresponding additional parameters and a more sophisticated surface layer model (Kaiser et al., 2011; Shiraiwa et al., 2010; Springmann et al., 2009). However, in the studies by Kahan et al. (2006) and Kwamena et al. (2004; 2007), the reactivity could be well described by a Langmuir – Hinshelwood

mechanism (i.e. one species [BaP] is adsorbed to the surface while the second species, ozone, is in phase equilibrium). This approach is followed in the CMAQ aerosol module by introducing the degradation rate coefficient k in the following form

$$k = \frac{k_{max} K_{O_3} c_{O_3}}{1 + K_{O_3} c_{O_3}} \tag{14}$$

where $k_{max}$ is the maximum rate coefficient (0.060 ± 0.018 s$^{-1}$), $K_{O_3}$ is the ozone gas to surface equilibrium constant (0.028 ± 0.014×10$^{-13}$ cm$^3$), and $c_{O_3}$ the concentration of ozone (molecules/cm$^3$). As this reaction neglects the substance fraction not

accessible for gaseous O$_3$ molecules (so-called burial effect; Zhou et al., 2012), an upper estimate for BaP reactivity is simulated.

## 2.4 Dry and wet deposition

SOCs and their degradation products are removed from the atmosphere through dry and wet deposition. Dry deposition is modelled in CMAQ as a one-way flux out of the lowest layer of the atmosphere. Particulate forms have dry deposition

velocities determined by Brownian diffusion, turbulence, and sedimentation. Mass and diameter of the mode are used to estimate the sedimentation velocity (Binkowski, 2003; Byun and Ching, 1999). The deposition velocity of gaseous molecules is based on an electrical resistance analogue that includes aerodynamic ($R_a$), quasi-laminar boundary layer ($R_b$), and surface canopy ($R_c$) resistance (Pleim et al., 1999). $R_a$ is computed using similarity theory with heat flux. $R_b$ depends on landuse specific friction velocity and molecular characteristics of gases. $R_c$ is consisted of several components, including

bulk stomatal, dry cuticle, wet cuticle, ground, and in-canopy aerodynamic resistance (Pleim et al., 1999). Cuticle and ground resistances are based on ozone and sulphur dioxide observations and neglect absorption into organic plant or soil material. Values for time-dependent $R_c$ parameters are determined via landuse and mesoscale meteorological models, which provide fractional vegetation cover, leaf area index, fractional leaf area wetness, and leaf stress associated with radiation, root zone soil moisture, temperature, and humidity (Pleim and Xiu, 1995).

Wet deposition of gaseous molecules occurs through in-cloud and below-cloud scavenging. Both processes are assumed to have the same scavenging efficiency and dependency on Henry's Law, water content of the cloud, and precipitation rate (Chang et al., 1986). Wet scavenging parameterization of the particulate phase molecules follows the method of Roselle and Binkowski (Byun and Ching, 1999) and depends on the aerosol mode (size) and time required to remove all liquid water from a cloud volume. No ice phase scavenging is included in the 4$^{th}$ version of the aerosol module in CMAQ.

**3 Model implementation and study design**

The model domain included entire Europe and neighbouring countries as shown in Figure 1. The domain is within the area covered by the UNECE-CLRTAP (and its POP protocol, covering PAHs) and was specifically chosen such as to exploit recently prepared high-resolution emission data (Bieser et al., 2011) and make use of the only monitoring network in the region (EMEP, see below). Computational simulations were performed on a 36 × 36 km Lambert Conformal Conic grid

using offline hourly meteorological fields, generated by the Weather Research Forecasting (WRF) model version 3.4.1 using GFS meteorological reanalysis data (spatial resolution $0.5° \times 0.5°$) as input. The parameterization of WRF included the following schemes: Milbrandt-Yau Double-Moment 7-class microphysics (Milbrandt and Yau, 2005a; 2005b), Rapid radiative transfer model (RRTMG) longwave and shortwave scheme (Iacono et al., 2008), asymmetric convective model of PBL (Pleim, 2007), Pleim-Xiu surface layer model (Pleim and Xiu, 2003; Xiu and Pleim, 2001), and the improved version of Grell-Devenyi (Grell, 1993; Grell and Devenyi, 2002) ensemble scheme for cumulus parameterization. On the vertical dimension, the domain was resolved in 36 layers, following the eta coordinate system. Subsequently, CMAQ-ready meteorological input fields for 2006 were prepared using the Meteorology-Chemistry Interface Processor (MCIP) (Otte and Pleim, 2010). Computational simulations were performed in a distributed computing infrastructure operated and managed by the Czech National Grid Organization MetaCentrum NGI.

### 3.1 Emissions

Capturing the spatiotemporal pattern of emissions is of crucial importance to determine the fate and transport of PAHs. Several attempts have been carried out during the past decades, with varying spatial and temporal resolutions in historical, current, and future projection terms for Europe (Bieser et al., 2012; Pacyna et al., 2003). Currently, 4 database sources with gridded BaP emissions exist for Europe and Russia: the TNO database (Denier van der Gon et al., 2006), the EMEP one (Mantseva et al., 2004), a dataset for POPCYCLING-Baltic project (Pacyna et al., 2003), and a global emission inventory for the year 2008. The SMOKE for Europe (SMOKE-EU) model (Bieser et al., 2011) output used in this study, follows a methodology that is based on the TNO database and has been evaluated in a number of subsequent CMAQ applications (Aulinger et al., 2011; Bewersdorff et al., 2009; Matthias et al., 2008). In order to spatially disaggregate BaP emissions in a more realistic way, SMOKE-EU improves on using a linear dependency on the population density as surrogate factor, by introducing different relevant relationships based on source classification and meteorological parameters. For example, the relationship to wood availability and the level of urban agglomeration have been introduced as an effort to include the effect of burning biomass. The resulting spatial distribution and annual average emission flux of BaP for our domain during the year 2006 is shown in Figure 1. In addition, BaP emissions are shaped temporally using diurnal and weekly functions. As residential combustion is associated with the majority of BaP emissions, a strong dependence on the season and spatial variability is expected. To account for this dependence, heating degree days have been introduced as a proxy (Aulinger et al., 2011). A major strength of the SMOKE-EU inventory is provision of CMAQ-ready emissions for all species involved in the desired chemical mechanism (e.g. CB5) and aerosol module (e.g. aero4). Following the trimodal representation of aerosol in CMAQ, 0.1% of the emissions of BaP was allocated to the Aitken mode, while the rest was emitted in the accumulation mode, as this is the default treatment with all primary organic aerosol species in the model.

### 3.2 Evaluation data and sensitivity simulations

The spatiotemporal coverage of BaP in air and deposition in Europe and Russia is limited, in many regions sporadic or not available at all. For 2006, concentration in air is available from eight stations (n=173), while deposition information is available from six stations of the EMEP monitoring network (Aas and Breivik, 2012). An overview of the data, including site codes, geographical information, and the sampling strategies is provided in Table S2. It is evident that during 2006, only four of the total of ten sites provide information on concentration and deposition simultaneously. Additionally, the sampling strategy varies widely among the stations, leading to a non-homogeneous pool of data. For example, on the Košetice (CZ0003R) and Niembro (ES0008R) stations, BaP is measured for 24 hours once a week, while in Aspvreten (SE0012R) and in Pallas (FI0036R) samples cover one week and are taken once every month. On the other hand, the Latvian sites (LV0010R, LV0016R) are sampled on a monthly basis, while for the High Muffles location (GB0014R) a three-month sample is obtained four times in 2006. Finally, the sites located in Germany (DE0001R, DE0009R) provided information only in terms of wet deposition and on a monthly basis during 2006.

Based on the set of GPP mechanisms described before, the effect of each parameterization was evaluated against the EMEP measurements by incremental testing of the models involved, while keeping the same meteorological and emission inputs. Table 2 summarizes the 5 annual scenarios that were developed for 2006 (1: Junge-Pankow, 2: Scenario 1 with water dissolution, 3: Scenario 2 with the Harner-Bidleman (HB) $K_{oa}$ absorption model, 4: Scenario 2 with dual OM absorption and black carbon adsorption model Dachs-Eisenreich (DE) scheme, 5: Scenario 4 supplemented by a heterogeneous reactivity module). A spin-up time of 7 days was allowed for each simulation in order to maintain a low influence of the initial conditions. Comparisons against measurements of BaP in ambient air were produced using the statistical computing framework of the R language, and the model performance was evaluated building on a wide range of statistics calculated using the Models-3 and the openair packages (Carslaw and Ropkins, 2012).

## 4 Results and discussion

### 4.1 Spatiotemporal distributions of BaP over Europe

The pattern of normalized annual mean BaP concentrations over the domain using the baseline scenario 1 (JP) is presented in Figure 2a. Elevated levels were found in regions with strong emission sources in Central Europe (Po Valley, Rhine-Ruhr, Poland and Ukraine areas), as well as in the vicinity of large cities (among others Moscow, Paris, Vienna, Madrid, and Istanbul). In addition, annual mean BaP concentrations according to the fully expanded scenario 5 (KW) are depicted in Figure 2b. Comparing the output from the aforementioned simulations revealed higher BaP levels with scenario 5, which can be associated with the geographic distribution of emissions. This is the first indication that the chosen partitioning approach has a significant impact on the model results for total BaP concentrations. However, it is also evident that in southern Europe and the Mediterranean region the BaP losses associated with the addition of an upper estimate of heterogeneous reactivity

with O$_3$ (Scenario 5 – Figure 2b) may have a limiting effect on BaP long-range transport (LRT) potential. This can be supported by the prominent BaP gradients calculated over the Mediterranean Sea, acting as a sink that is driven by low BaP emissions and elevated O$_3$ levels. The overall model patterns show a good agreement with recent regression methods mapping BaP levels that provide with a partial coverage of our EU domain. With respect to previous studies involving simulations of BaP over Europe we find lower BaP concentrations, but nevertheless within the same order of magnitude of those estimated by initial CMAQ applications (Aulinger et al., 2007; Bewersdorff et al., 2009; Matthias et al., 2009). This supports the perception that BaP emissions over Europe have decreased substantially during the recent decades, a feature improved in subsequent evolution of the emissions model which led to better agreement with the more recent CMAQ applications (Bieser et al., 2012). A closer look over Italy and the Iberian peninsula reveals a good agreement with previous studies depicting BaP distributions for these regions (San José et al., 2013; Silibello et al., 2012). Finally, discrepancies over Eastern Europe indicate that a more elaborate emissions inventory (i.e. domestic heating adjustments, fire activities) would be beneficial to address the elevated BaP levels predicted by recent hybrid modelling approaches (Guerreiro et al., 2016a).

### 4.1.1 Effect of GPP parameterizations and heterogeneous reactivity with O$_3$

Figure 3 and Table 3 illustrate the differences resulting from the incrementally tested scenarios (Table 2), expressed as annual mean BaP concentrations over the entire domain of interest and the individual EU-28 member states. As expected, the effect of dissolution into aerosol water (Figure 3a) is limited throughout most of the domain, with coastal urban areas being the most influenced (i.e. Istanbul, North Sea). BaP dissolution in aerosol liquid water according to scenario 2 led to less than 0.36% reduction of ground level BaP in Europe as compared to the baseline JP scheme (Table 3). Introducing the HB absorption scheme (scenario 3) resulted in ~119.2% higher average BaP concentrations (Figure 3b) with extremes observed in areas associated with emission sources in Central Europe (i.e. Po Valley) and large cities (i.e. Moscow, Paris, Vienna, and Istanbul). Expressed in terms of domain-wide atmospheric lifetimes (defined as $\tau = b_{tot}/F_{em}$, where $b_{tot}$ denotes the total atmospheric burden and $F_{em}$ the total emissions), scenario 3 leads to a 210% increase for BaP over scenario 2. Similarly, shifting to the DE scheme (scenario 4) appears to have an additive effect on near-ground BaP concentrations with total lifetimes reaching a domain-average increase of 13%. Compared to the scenario 3, scenario 4 affects a larger area of the domain (Figure 3c), even though the magnitude of the effect is less pronounced due to its association with the availability of black carbon. The domain mean of modelled elemental carbon is 0.0212 μg m$^{-3}$ and the spatial distribution is matching its emission pattern (Figure S3). These levels are significantly lower compared to field studies in the urban air of south-west Europe (EEA, 2013; Milford et al., 2016), and may subsequently lead to hindered performance of the DE scheme.

Finally, the introduction of heterogeneous chemistry (scenario 5) had, as expected, a substantially negative effect on the BaP concentrations within our domain culminating in major conurbations (i.e. Po Valley, Moscow, and Istanbul; cf. Figure 3d). Heterogeneous degradation due to the reaction with O$_3$ accounts for an approximate 18.8% reduction of the mean ground level BaP and a 25% reduction of its atmospheric lifetime in the European domain (Table 3). As mentioned in Sect. 4.1, the BaP concentration gradients reveal a strong spatial link with emission sources in the south, as well as high O$_3$ concentrations

typically found across the coasts of the Mediterranean Sea (Figure S4). CMAQ calculates ozone patterns that are in accordance to the findings of recent analyses, exhibiting similar performance compared against measurements at the European level (de Smet et al., 2009).

### 4.1.2 Seasonality and relevant parameters affecting BaP fate and transport

Seasonally disaggregated distributions of BaP, as simulated scenario 4 are presented in Figure 4. Strong emissions during wintertime and fall have an apparent effect on the average ground level concentrations. In addition, a normalized effect of long-range transport for BaP can be observed in the vicinity of Istanbul and other coastal metropolitan areas, over parts of the large water bodies of the Mediterranean and the Black Sea. This scenario, however, does not include the treatment of the heterogeneous reaction with ozone, and therefore provides an upper bound estimate of the LRT potential of BaP. On the
other hand, Figure 5 demonstrates the seasonal effect of the heterogeneous reaction of BaP with ground-level ozone on BaP concentrations. Distributions of BaP concentration differences reveal a stronger reduction during winter that is influenced by central and southern European emission sources and mid-latitude ozone availability. Interestingly, the differences between winter and summertime appear to be comparable in terms of magnitude, despite the substantially weaker sources during summer that shift the process to the south.

An important factor for the performance of the BaP model is the mass concentration and composition of the aerosols in the three different modes and their relationship with the temperature of the ambient air. Figure 6 illustrates the aerosol distributions as simulated at model cells representing three different settings (Urban: City of Vienna, Suburban: CZ0003R, Remote: FI0036R) during a typical winter and summer day. As we move further away from the vicinity of the sources, we notice not only a decrease of the overall BaP levels but also a change in the distribution influenced by each individual
simulated scenario. While the basic Junge-Pankow implementation results in higher coarse mode BaP concentrations, the HB and DE schemes accentuate the strength of the accumulation mode where most of the BaP aerosol mass is emitted. The disproportionately higher coarse BaP fractions calculated at the remote grid cells during the summertime can be explained by conditions favouring long-range transport.

Regarding the parameterizations of the HB and DE schemes, the calculation of the parameter $K_{oa}$ has a direct effect on the
relevant model results (Scenarios 3 and 4). Use of the regression parameters for the semi-logarithmic temperature-dependent form proposed by Odabasi et al., (2006), results in a mixed effect over its predecessor scheme that is seasonally disaggregated and presented in Figure S5. These figures suggest that the partitioning based on each $K_{oa}$ approach is sufficiently similar for most of the domain during the winter. However, differences between the two expressions with respect to simulating summertime BaP concentrations over Central Europe were noted. In addition, simulations that address the
uncertainties contributed by the method of estimating $K_{SA}$ were performed for the winter period of 2006 using the DE scheme. Figure S6 reveals similar results with BaP levels slightly higher in cells with strong emissions sources. However, with the exception of Moscow, the average difference BaP concentration in urban cells did not exceed 0.02 ng m$^3$ when $K_{SA}$

was calculated based on the thermodynamic estimation model suggested by van Noort (2003). Similarly to $K_{oa}$, results suggest that the partitioning based on each $K_{SA}$ approach is sufficiently similar.

## 4.2 Model evaluation against EMEP measurements

The EMEP network database and model output from all 5 scenarios were imported in the R language framework to create measurement-model pairs for further analysis. Figure 7 illustrates the resulting timeseries plots for selected EMEP sites against CMAQ output extracted from scenario 4. This figure also demonstrates differences in terms of sampling protocols among individual sites, with the extreme of the High Muffles station (4 samples/year – Table S2). Aggregate temporal profiles of the modelled and measured BaP concentrations across all sites depicted in Figure 8 reveal a strong seasonal pattern that is captured well by the model. However, BaP concentrations appear to be underestimated during the colder months. This effect can be attributed to the strength of emissions, which is probably related to inadequate coverage of residential heating in current inventories (wood/biofuel burning). Figure S7 illustrates this by comparing annual average BaP emission fluxes between model cells enclosing EMEP monitoring site locations, and selected metropolitan areas of Europe during 2006. While known deficiency of PAH inventories, it may limit the usefulness of background measurements (such as those in the EMEP network) in assessing the overall model performance under different GPP scenarios.

It is evident from the metrics obtained across all sites (Table 4 and Taylor diagram summary Figure S8) that the increased complexity in GPP formulation results in better agreement with the EMEP measurements. For reasons mentioned in Sect. 2.2-2.3 and 4.1.1, deviations from this observation are the simulated scenarios that introduce the dissolution to aerosol water (scenario 2) and degradation due to the heterogeneous reaction with ozone (scenario 5). Overall, the dual model (DE) employed in scenario 4 showed the best agreement with the measurements. The DE parameterisation also performed best when compared with the JP and HB schemes on global scale applications (Lammel et al., 2009). However, when looking at site-specific performance metrics (Table S9), the Košetice location (CZ0003R) reveals a better agreement with the fully expanded scenario 5 that treats degradation by ozone. The Košetice site is the closest to conurbations among the EMEP sites available during 2006. This could indicate that the heterogeneous chemistry, which is implemented as an upper estimate in this model, tends to underestimate BaP levels further away from the sources. The best model performance was observed at the Pallas station (FI0036R), where scenario 4 was capable of matching 50% of the measurements within a factor of 2. For the remaining remote sites in Finland, the simulation under scenario 5 calculated underestimated BaP levels. Based on the Index Of Agreement (IOA) metric, a ranking of the overall model performance similar to the Pallas site was observed for the rest of the monitoring sites. Comparing site-specific performance against previous modelling efforts over Europe (Aulinger et al., 2007; Matthias et al., 2009), we reach the same conclusion pointed out with the spatial distributions, i.e. we attribute the lower performance metrics presented here to the substantially weaker emission sources in recent years, hence the better agreement with the more recent SMOKE-EU/CMAQ application (Bieser et al., 2012). With respect to other annual simulations, we observed lower average BaP levels over the Piedmont region compared against the FARM model but also higher than the EMEP (MSC-E) output used in the same study (Silibello et al., 2012). Finally a qualitative analysis of the

deposition measurements for the few stations that measured BaP showed that CMAQ calculated plausible patterns of dry and wet deposition (cf. Figure S10).

## 5 Conclusions

This study presented the development of a new modelling system for investigating the dynamics of transport and gas-particle partitioning schemes for POPs at the regional scale over Europe. The implementation of this model is based on the WRF-CMAQ model framework with additional algorithms for GPP schemes and a module that accounts for heterogeneous reactivity of BaP with ozone. Predictions from the WRF-CMAQ-BaP model were compared against measurements obtained from the EMEP monitoring network. Model evaluation has revealed satisfactory agreement to the measurements and performance metrics similar to those of previous studies with significantly higher measurement availability. The results presented in this work suggest that the new expanded model is able to simulate fairly well the ambient levels of BaP. It is found that BaP distributions in Europe are very sensitive to the choice of GPP models. While this statement has been also indicated at the global scale (Lammel et al., 2009), GPP has not been adequately explored by previous regional scale studies (cf. Table S1 and references therein). We conclude that the dual OM absorption and black carbon adsorption (DE) parameterisation (scenario 4-5) offer a better performance, as BaP predictions tend to be closer to the measurements. Introducing an independent upper estimate of heterogeneous BaP reactivity is found to significantly reduce BaP levels throughout the domain, with a reduction pattern that follows the spatial distribution of ozone and emission sources. Despite the limited BaP monitoring network, a better agreement was observed when complementing WRF-CMAQ-BaP with the upper estimate for heterogeneous reactivity (scenario 5) at certain EMEP sites. Uncertainties and limitations of the current emission inventory estimates of BaP, OM, and BC along with insufficient laboratory data are of major concern, hampering the efforts to fully evaluate the long-range transport potential and the effect of photochemistry and interactions of such compounds. In addition, certain simplifications in the treatment of coarse PM and OM within the aerosol module of CMAQ (Binkowski, 2003; Mathur et al., 2008) are posing limitations to fully exploit the advanced empirical gas-particle partitioning models ($K_{oa}$, Dachs-Eisenreich) in this specific mode. An implicit overlap of the empiric adsorption model (JP) with the $K_{oa}$ (or another absorption) model (see section 2.1.1) may artificially enhance partitioning, something that has not been addressed by previous modelling efforts. However, such discrepancy is expected to be negligible for BaP, especially when compared to existing uncertainties of output aerosol parameters propagating from emission estimates and from the current design and state of development of the CMAQ aerosol module (AERO).

The modelling approach presented here allows simultaneous estimation of organic pollutants, including semivolatiles (such as most POPs) within the CMAQ framework, and can be used as a supplementary component of a population exposure modelling system. Although similar systems are currently explored, they rely heavily on observation data while using a very basic air quality model structure (Guerreiro et al., 2016a; 2016b). At this time, annual exposure estimates are more useful due to the low accuracy accompanying highly time-resolved models and observations. As previous efforts to evaluate the use

of mesoscale models for BaP dispersion studies also conclude, changes in emission patterns and their strength is of particular importance. While there is undeniable effort to shift from fossil fuels to renewables which has led to updated emission inventories, domestic heating across Europe has also a complex relationship to economic factors and human activities that varies tremendously between countries and fuel types (Lohmann et al., 2006; Saffari et al., 2013). Besides the need of updated emission inventories for the Eastern Europe, extended simulations are suggested during years with higher measurement availability and density. Following modelling studies should focus on quantifying the long-range transport potential and examining the hypothesis that secondary organic aerosol (SOA) may facilitate PAH transport by utilising more sophisticated aerosol and heterogeneous chemistry parameterisations or submodels (i.e. accounting also for the burial effect - cf. Sect. 2.2). More specifically, recent evidence suggests that in cold and dry air accessibility of PAHs in OM is reduced (due to low diffusivity), which might explain the apparent inconsistency of high LRT potential (BaP levels in the Arctic) on one hand side and relative high heterogeneous reactivity measured in the laboratory (Friedman et al., 2014; Zelenyuk et al., 2012; Zhou et al., 2012; 2013). As understanding of PAHs' atmospheric lifetimes, PAHs' interaction with SOA, and chemical composition of ambient OM progresses, a new need for additional studies quantifying GPP and LRT potential under a wide range of atmospheric conditions is emerging. In view of such findings, the WRF-CMAQ-BaP modelling system should be extended to study a wide range of additional organic pollutants and processes (i.e. multicompartmental cycling, biodegradation, heterogeneous chemistry).

**Acknowledgements**

The authors thank Frank Binkowski, Sarav Arunachalam, and Zac Adelman at the University of North Carolina (UNC) for their guidance and support during the onset of this project. EMEP is acknowledged for the monitoring data. Access to computing and storage facilities owned by parties and projects contributing to the National Grid Infrastructure MetaCentrum provided under the programme "Projects of Large Infrastructure for Research, Development, and Innovations" (LM2010005) is also greatly appreciated. This research was supported by the Granting Agency of the Czech Republic (project No. 312334), the National Sustainability Programme of the Czech Ministry of Education, Youth and Sports (LO1214), and the RECETOX research infrastructure (LM2015051).

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

Table 1. Physicochemical parameters and reaction rate coefficients used in the WRF-CMAQ-BaP model.

| | | |
|---|---|---|
| Inverse Henry's Law Constant[1] | $\log H'$ | -4.7 |
| Subcooled liquid vapour pressure[1,2] | $\log p_L$ | -5.2 (Pa) |
| Octanol-Water partition coefficient[1] | $\log K_{ow}$ | 5.9 |
| Octanol-Air partition coefficient[1,3] | $\log K_{oa(dry)}$ | $11.1^1$ (b=12.04, m=5382)[3] |
| Water saturated octanol-Air partition coefficient[1] | $\log K_{oa(wet)}$ | 10.6 |
| Diesel Soot-Water partition coefficient[4] | $\log K_{soot-water}$ | 8.4 |
| Diesel Soot-Air partition coefficient[4,8] | $\log K_{soot-air}$ | $13.0^4$, $11.59^8$ |
| Gas-phase OH reaction rate constant | $k_{OH}$ | $50 \times 10^{-12}$ ($cm^3$ $molec^{-1}$ $s^{-1}$) |
| Ozone gas-surface equilibrium constant[6] | $K_{O3}$ | $0.028 \times 10^{-13}$ ($cm^3$) |
| Maximum rate coefficient[6] | $k_{max}$ | 0.060 ($s^{-1}$) |

[1](Beyer et al., 2000); [2](de Maagd et al., 1998; Offenberg and Baker, 1999); [3](Odabasi et al., 2006), $\log K_{OA} = m/T(K) + b$; [4](Bärring et al., 2002), $\log K_{soot-water} = 1.39 \log K_{ow} + 0.1$; [5]($\log K_{soot-air} = \log K_{soot-water} - \log H'$); [6]Estimated values (EPIWIN;USEPA); [7](Kwamena et al., 2004), [8](van Noort, 2003)

5    [1]

Table 2. Summary of the simulated scenarios designed to test gas-particle partitioning and heterogeneous reactivity of BaP.

| Description and references | Scenario | | | | |
| --- | --- | --- | --- | --- | --- |
| | 1 | 2 | 3 | 4[*] | 5 |
| Gas phase reaction with OH | Yes | Yes | Yes | Yes | Yes |
| Adsorption model (JP)[1] | JP | JP | JP | JP | JP |
| Aerosol water dissolution (W) | - | W | W | W | W |
| Absorption to OM (HB)[2] | - | - | HB | - | - |
| Absorption to EC/BC (DE)[3] | - | - | - | DE | DE |
| Heterogeneous reaction with $O_3$ (KW)[4] | - | - | - | - | KW |

[1]Junge, 1977 and Pankow, 1987; [2]Harner and Bidleman, 1998; [3]Dachs and Eisenreich, 2000; [4]Kwamena et al., 2004; [*]Additional simulations with temperature-dependent octanol-air partition coefficient calculated based on Odabasi et al., 2006 and a thermodynamic estimation model suggested by van Noort (van Noort, 2003).

Table 3. Average ground level BaP concentrations, $c_{BaP}$ (pg/m$^3$) over the entire domain and in individual EU-28 member states for all simulated scenarios, and concentration changes $\Delta c_{BaP}$ (%) between subsequent scenarios.

| | 1 | 2 | | 3 | | 4 | | 5 | |
|---|---|---|---|---|---|---|---|---|---|
| | $c_{BaP}$ pg/m$^3$ | $c_{BaP}$ pg/m$^3$ | $\Delta c_{BaP}$ % | $c_{BaP}$ pg/m$^3$ | $\Delta c_{BaP}$ % | $c_{BaP}$ pg/m$^3$ | $\Delta c_{BaP}$ % | $c_{BaP}$ pg/m$^3$ | $\Delta c_{BaP}$ % |
| Domain average | 10.4 | 10.4 | -0.4 | 22.8 | 119.2 | 28.3 | 24.1 | 23.0 | -18.8 |
| Austria | 58.4 | 58.4 | 0.0 | 89.0 | 52.4 | 103.4 | 16.1 | 86.5 | -16.3 |
| Belgium | 36.2 | 36.1 | -0.1 | 65.8 | 82.1 | 75.8 | 15.2 | 62.5 | -17.6 |
| Bulgaria | 14.7 | 14.7 | -0.1 | 28.9 | 97.1 | 37.7 | 30.4 | 26.7 | -29.3 |
| Croatia | 15.2 | 15.2 | -0.1 | 35.1 | 131.0 | 45.5 | 29.9 | 32.2 | -29.4 |
| Cyprus | 11.2 | 11.2 | 0.0 | 13.2 | 17.8 | 24.7 | 87.2 | 10.1 | -59.1 |
| Czech Republic | 38.6 | 38.6 | 0.0 | 70.1 | 81.5 | 83.5 | 19.1 | 67.9 | -18.7 |
| Denmark | 7.3 | 7.2 | -1.3 | 22.4 | 213.0 | 26.0 | 16.0 | 24.5 | -5.7 |
| Estonia | 9.5 | 9.5 | -0.9 | 28.9 | 205.3 | 34.6 | 19.7 | 32.0 | -7.4 |
| Finland | 6.0 | 5.8 | -2.9 | 16.1 | 176.0 | 21.9 | 36.2 | 17.7 | -19.5 |
| France | 27.1 | 27.1 | 0.0 | 48.0 | 76.9 | 56.6 | 18.0 | 44.7 | -21.0 |
| Germany | 30.3 | 30.3 | -0.1 | 57.6 | 90.2 | 69.2 | 20.2 | 55.6 | -19.6 |
| Greece | 7.4 | 7.4 | -0.1 | 14.7 | 97.7 | 22.3 | 52.2 | 12.4 | -44.6 |
| Hungary | 29.3 | 29.3 | -0.1 | 61.3 | 109.5 | 75.1 | 22.5 | 59.7 | -20.5 |
| Ireland | 3.8 | 3.8 | -0.1 | 6.8 | 80.0 | 8.7 | 27.5 | 6.0 | -31.7 |
| Italy | 36.9 | 36.9 | 0.0 | 62.8 | 70.1 | 76.3 | 21.6 | 57.9 | -24.1 |
| Lithuania | 10.0 | 10.0 | -0.5 | 32.7 | 228.4 | 38.4 | 17.5 | 36.3 | -5.5 |
| Luxemburg | 37.1 | 37.1 | 0.0 | 62.0 | 67.2 | 71.7 | 15.8 | 57.9 | -19.3 |
| Latvia | 6.0 | 5.9 | -1.3 | 25.7 | 335.0 | 30.2 | 17.7 | 29.8 | -1.4 |
| Malta | 0.5 | 0.4 | -0.5 | 0.5 | 20.6 | 0.7 | 37.1 | 0.5 | -36.6 |
| Netherlands | 24.0 | 23.9 | -0.3 | 52.5 | 119.5 | 63.1 | 20.2 | 51.5 | -18.4 |
| Poland | 25.6 | 25.6 | -0.2 | 56.7 | 121.7 | 67.4 | 18.8 | 57.4 | -14.8 |
| Portugal | 15.0 | 15.0 | 0.0 | 20.9 | 39.1 | 28.4 | 36.3 | 18.1 | -36.5 |
| Romania | 12.7 | 12.7 | 0.0 | 27.4 | 115.8 | 36.0 | 31.4 | 25.7 | -28.6 |
| Spain | 8.4 | 8.4 | 0.0 | 11.9 | 41.8 | 16.8 | 41.4 | 9.3 | -44.8 |
| Sweden | 3.7 | 3.6 | -2.4 | 11.3 | 211.7 | 14.3 | 26.7 | 13.2 | -7.7 |
| Slovakia | 27.0 | 27.0 | -0.1 | 53.5 | 98.1 | 65.2 | 21.9 | 52.5 | -19.5 |
| Slovenia | 28.5 | 28.5 | 0.0 | 60.3 | 111.3 | 71.3 | 18.3 | 57.0 | -20.1 |
| United Kingdom | 3.3 | 3.3 | -0.4 | 8.9 | 170.6 | 11.4 | 28.4 | 8.2 | -27.6 |

Table 4. Comparison between modelled and measured BaP concentrations for the entire 2006 EMEP dataset (n=173) along with performance metrics and ranking across all sites.

| Rank | Scenario | FAC2 | MB | MGE | NMB | NMGE | RMSE | r | COE | IOA |
|---|---|---|---|---|---|---|---|---|---|---|
| 1 | 4 | 0.29480 | -0.05622 | 0.07145 | -0.65264 | 0.82950 | 0.14729 | 0.35193 | 0.22443 | 0.61222 |
| 2 | 3 | 0.25434 | -0.06279 | 0.07280 | -0.72889 | 0.84516 | 0.15081 | 0.33423 | 0.20979 | 0.60490 |
| 3 | 5 | 0.21387 | -0.07021 | 0.07345 | -0.81511 | 0.85272 | 0.15179 | 0.47598 | 0.20273 | 0.60136 |
| 4 | 1 | 0.18497 | -0.07640 | 0.07887 | -0.88699 | 0.91559 | 0.16069 | 0.31088 | 0.14395 | 0.57197 |
| 5 | 2 | 0.17919 | -0.07649 | 0.07894 | -0.88798 | 0.91642 | 0.16076 | 0.30793 | 0.14317 | 0.57158 |

FAC: Fraction of predictions within a factor of two, MB: Mean bias, MGE: Mean Gross Error, NMB: Normalised mean bias, NMGE: Normalised mean gross error, RMSE: Root mean squared error, r: Correlation coefficient, COE: Coefficient of Efficiency, IOA: Index of Agreement.

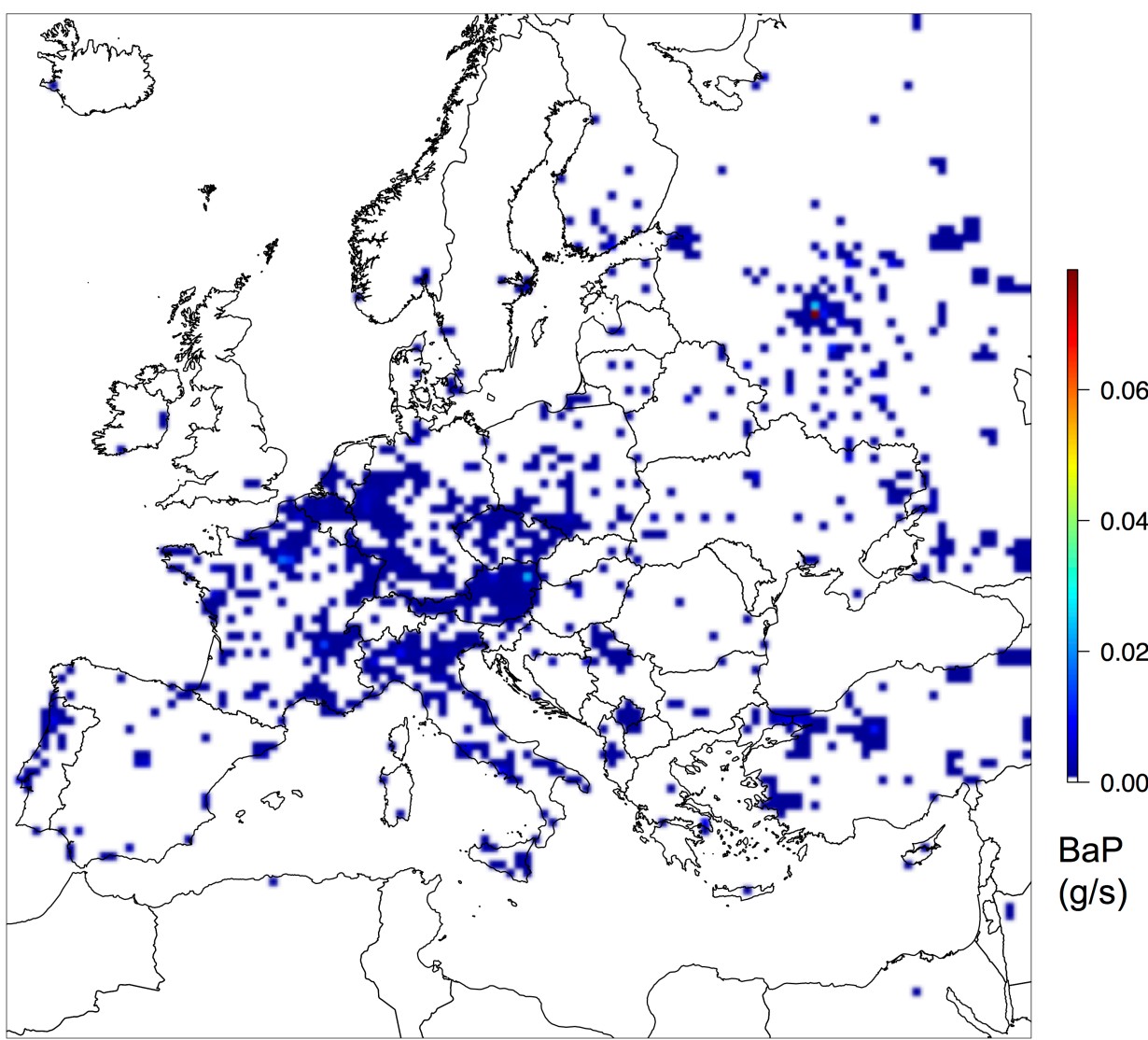

Figure 1. Study domain and 2006 annual grid cell-mean BaP emission flux (g s$^{-1}$) at the surface.

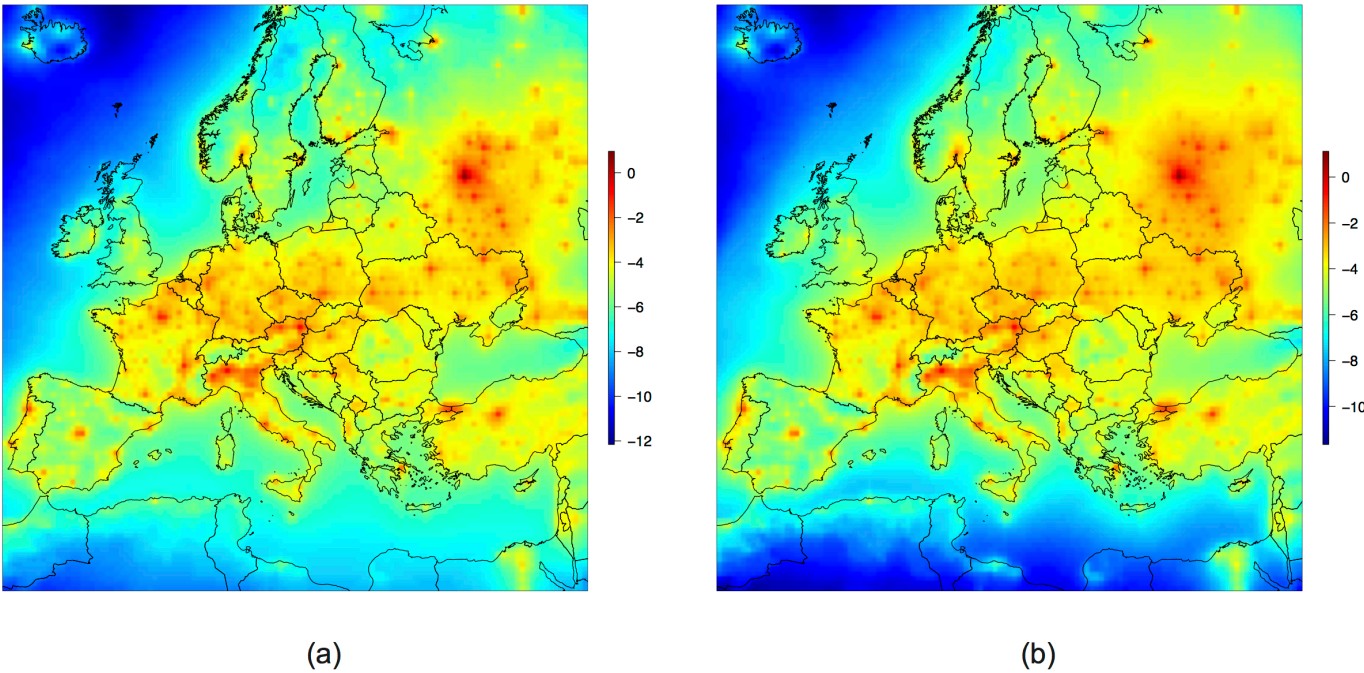

(a)              (b)

Figure 2. Annual average surface-level BaP concentrations [log $c_{BaP}$ (ng m$^{-3}$)] during 2006 plotted in logarithmic scale for (a) the Junge-Pankow (JP) scenario 1 and (b) for the fully expanded GPP scenario 5 that includes a heterogeneous reaction with ozone.

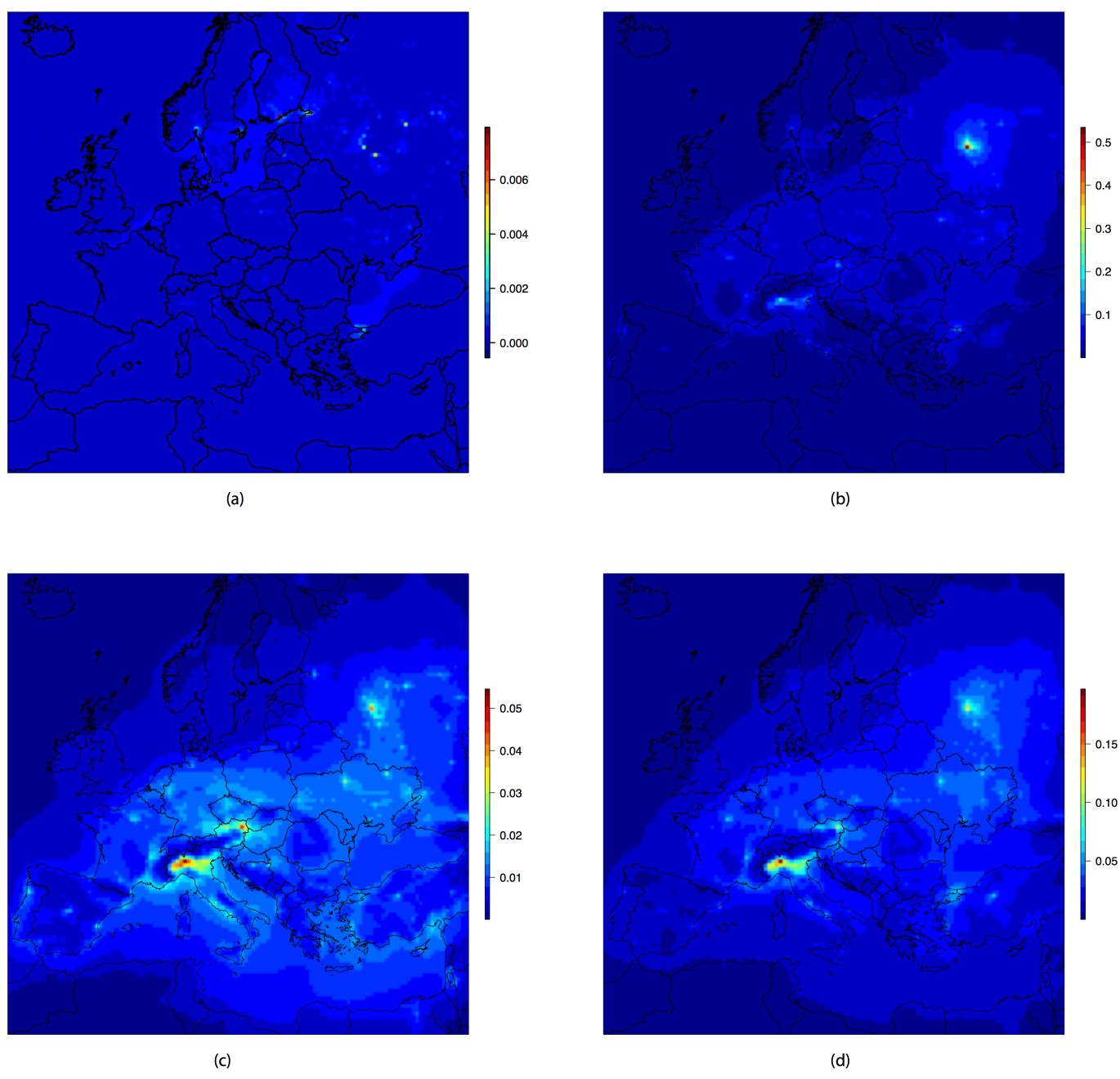

Figure 3. Differences in annual average surface-level BaP concentrations (ng m$^{-3}$) during 2006 plotted for (a) water dissolution effect: scenario 1 – scenario 2, (b) $K_{oa}$ effect: scenario 3 – scenario 2, (c) dual model effect: scenario 4 – scenario 3, and (d) $O_3$ degradation effect: scenario 4 – scenario 5.

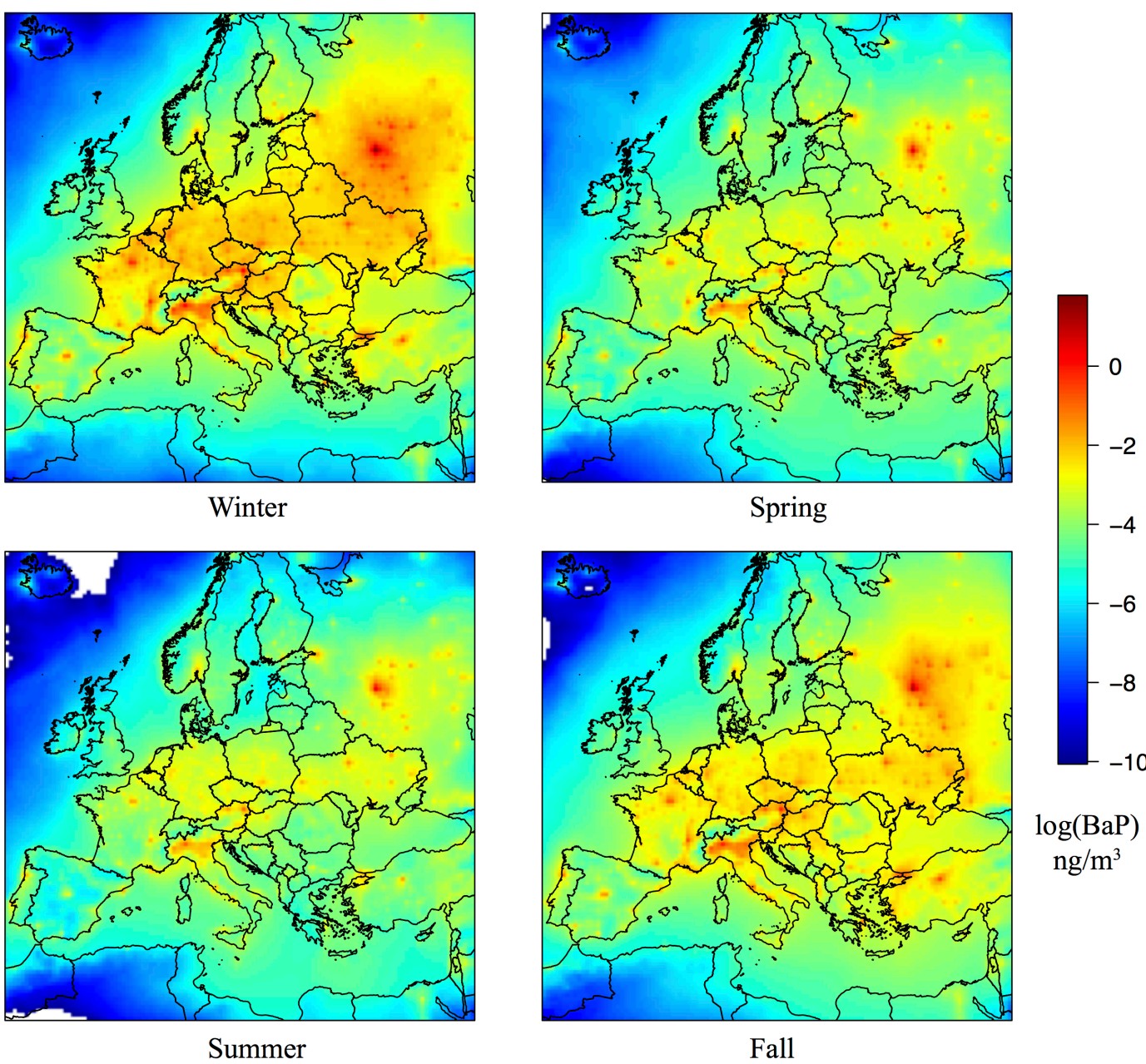

Figure 4. Seasonally disaggregated average surface-level BaP concentration [log $c_{BaP}$ (ng m$^{-3}$)] maps predicted by scenario 4 during 2006.

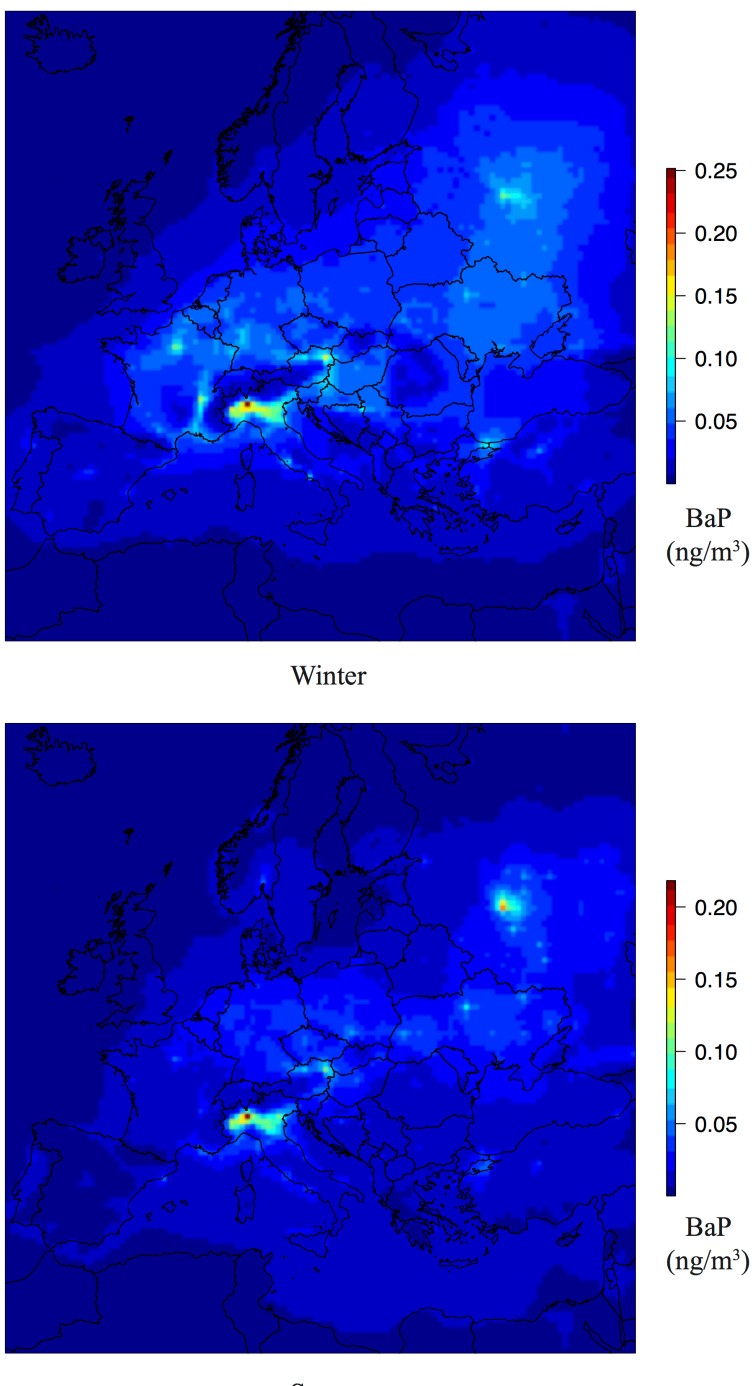

Winter

Summer

Figure 5. Average difference of surface-level BaP concentrations (ng m$^{-3}$) between scenario 4 and the fully expanded GPP scenario 5 that includes the heterogeneous reaction with ozone during the winter and summer months of 2006 (scenario 4 – scenario 5).

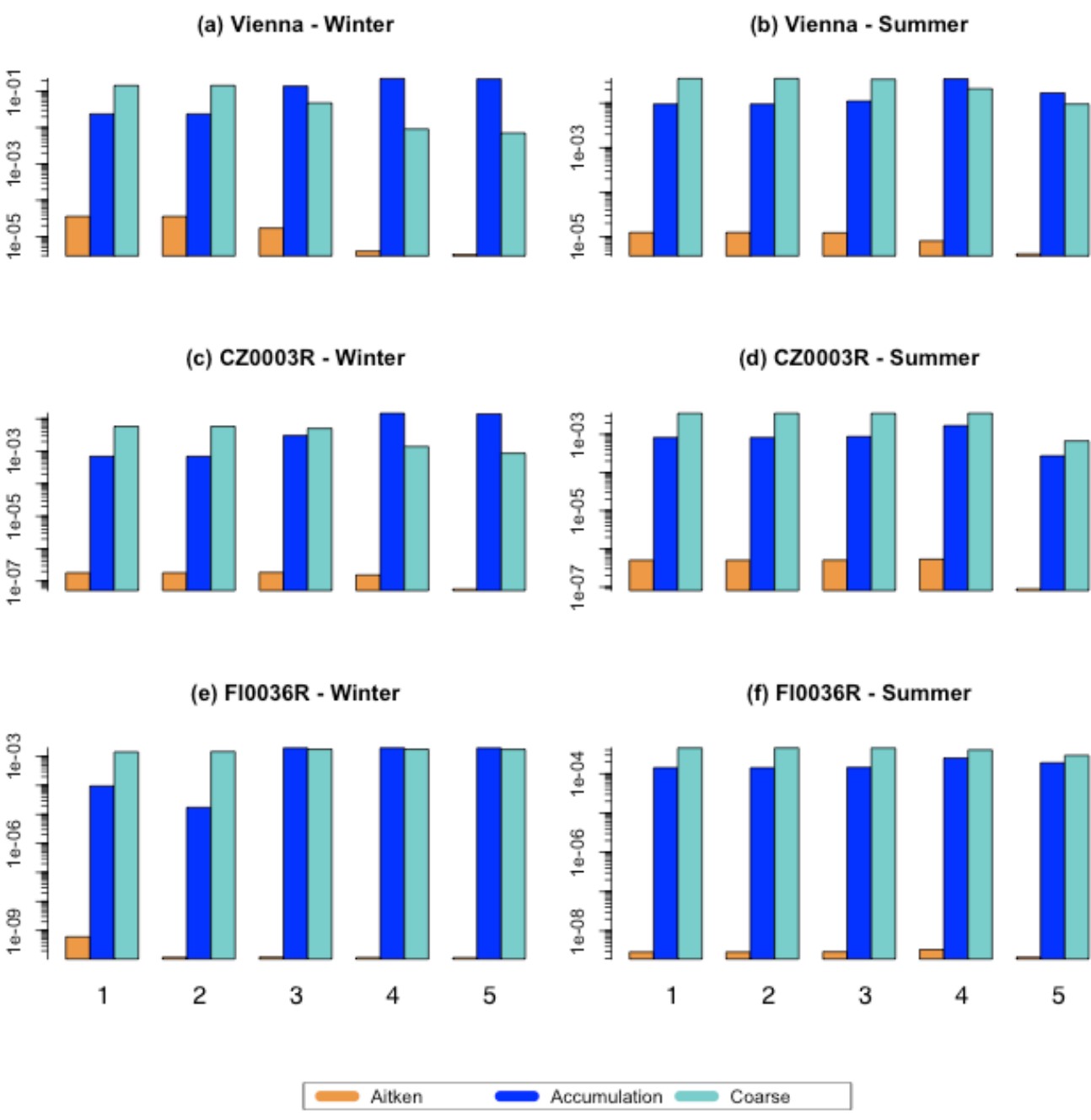

Figure 6. Distribution of BaP mass [log $c_{BaP}$ (ng m$^{-3}$)] in aerosols of three different modes in 3 different cells [Urban (Vienna), suburban (CZ0003R), and remote (FI0036R)] for each scenario as calculated during a winter (January 20) and a summer (July 20) day.

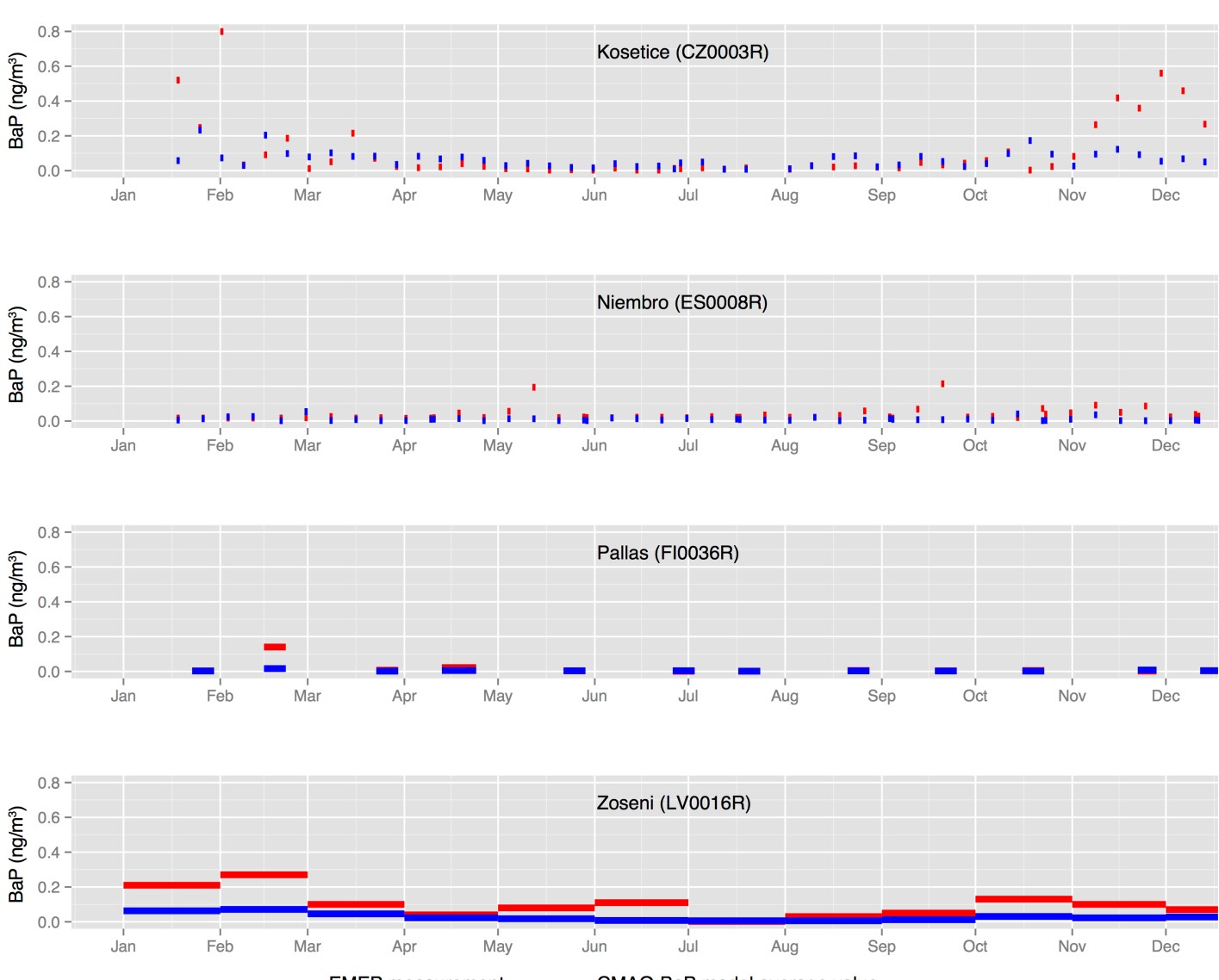

Figure 7. Model predicted and observed concentration (ng m$^{-3}$) timeseries at selected EMEP monitoring sites during 2006 for the dual GPP model scenario 4.

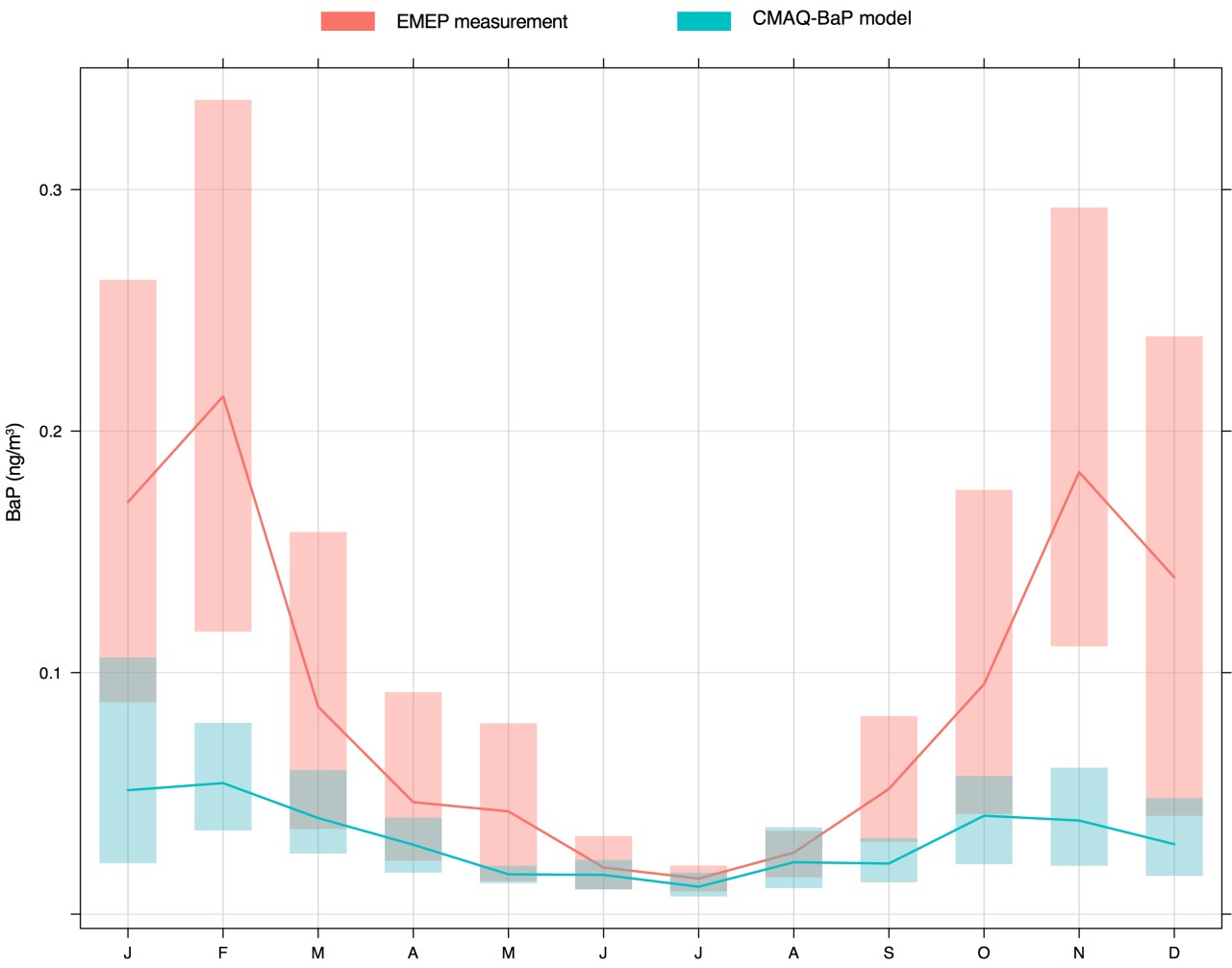

Figure 8. Model predicted and observed monthly BaP concentrations (ng m$^{-3}$) across all EMEP sites in 2006 for the dual GPP model scenario 4.

