# Peer review of "Evaluation of gas-particle partitioning in a regional air quality model for organic pollutants"

_Atmospheric Chemistry and Physics, 2016_

## Referee Comment (RC1) · Anonymous Referee #1 · 21 Mar 2016

This manuscript presents an evaluation of alternative gas-particle partitioning schemes in a modified version of the WRF-CMAQ model run for Benzo-a-pyrene (BaP). The authors compare modeled BaP concentrations within a European model domain under 5 different gas-particle partitioning schemes and also compare model results to measurement data from EMEP monitoring sites.

The novel aspect of this work seems to be announcing that the different gas-particle partitioning schemes have been incorporated into CMAQ. The more complicated schemes apparently provide the best model performance relative to measurement data, but the authors state that the disagreements between measurements and the model are likely mostly attributable to uncertainties in emission estimates. Therefore it is difficult to draw conclusions about which partitioning scheme is best based just on results reported in this paper.

[Figure]

I have two scientific concerns about the paper. I think both are significant, but possibly #1 arises from unclear presentation in the paper.

1) I understand Table 2 and the text to say that Scenarios 3, 4 and 5 include both the Junge-Pankow adsorption model *and* the Harner-Bidleman Koa absorption model. Is that correct? If so, I think the model is perhaps adding two redundant descriptions of gas-particle partitioning.

It has been a while since I looked at the JP model, but my recollection is that it includes a fitting parameter that was derived from empirical data under the assumption that all partitioning was by adsorption to aerosol surfaces. But, the assumption that adsorption dominates could not be confirmed. Xiao and Wania (http://www.sciencedirect.com/science/article/pii/S1352231003002139) later showed that vapor pressure and Koa both describe sorption to organic matter equally well. Therefore, I think partitioning schemes that include both vapor pressure and Koa as chemical parameters are likely including redundant information, and could easily be over-fitted.

2) The authors state that soot-air partition coefficients were calculated as the ratio of soot-water adsorption constants (Ksw) and the Henry's Law constant. Kai Goss has pointed out that applying a thermodynamic triangle for a solid interface in contact with water and the same solid interface in contact with an adjacent gaseous phase is a conceptual mistake that can lead to very large errors (http://pubs.acs.org/doi/abs/10.1021/es0301370).

I actually do not see a good option for the authors to overcome this problem, since the Lohmann & Lammel (2004) and the Dachs & Eisenreich (2000) papers that are the basis for the gas-particle partitioning scheme applied in this paper are both based on thermodynamic triangles that Goss argues (convincingly!) are invalid.

---

## Referee Comment (RC2) · Anonymous Referee #2 · 11 Apr 2016

General comments

The manuscript describes the application of community air quality model CMAQ to study atmospheric transport of B(a)P and effects of gas-particle partitioning and degradation. Several gas-particle partitioning models were implemented in the CMAQ model extending its capabilities with respect to modeling of PAHs. Though the subject of the study is of importance with regard to existing air quality problems in Europe, description of this study suffers from obscurity and thus requires substantial revisions.

First of all, I would mention not clear enough description of parameters applied in the equations 10 and 11, related to gas-particle partitioning (e.g. KOA, KSA). Secondly, evaluation of the effects of several GPP mechanisms through their incremental testing (incremental testing is defined in section 3.2 and Table 2) is not quite clearly described

with regard to sensitivity of CMAQ model output to particular partitioning mechanisms in model simulation scenarios. In some cases abbreviations, defined for GPP models and scenarios and used throughout the analysis, are mixed like e.g. DE model and DE scenario. This leads to problems with understanding what is presented, for example, in the second column of tables 4 and S8 called 'Model'. There are both abbreviations for scenarios and models (for example, JP-W and DE, HB, etc.). Finally, the underestimation of observed B(a)P concentrations in air is explained by low emissions. At the same time, there is no discussion of potential problems of emission data used, no comparison of total annual emissions with other studies, for example, with similar studies e.g. Aulinger et al. (2007) and others.

Specific comments

Page 3, line 12-13: "CMAQ contains modules representing advection, eddy diffusion, in-cloud, and precipitation processes". It would be better to use 'in cloud and below-cloud scavenging with precipitation'.

Page 3, line 26: 'gas phase reactions' instead of 'gas reactions'.

Page 4, line 23: Though the equation 2 follows the publication of Aulinger et al. (2007), it is not clear how the particulate fractions in each mode i, $f_i$, are obtained, because in the cited work (Cooter and Hutzell, 2002) similar equation is written for the sum of partition coefficients, but not for fractions of a compound in particulate phase.

Page 5, equation 7: $f_{OM}$ is used without index i. Does it mean that the fraction of organic matter in aerosol particles is the same in each of three modes?

Page 6, line 3: assumptions used here for ratio of activity coefficients and ratio of mean molar weights of organic matter of the particles and octanol (that they are equal unity) need to be discussed with regard to their uncertainties as it was shown in e.g. (doi: 10.5194/acpd-14-21341-2014 ).

Page 6, line 14-15: Soot-air partition coefficients were calculated as the ratio of sootwater adsorption constants KSW and the inverse Henry's Law constant (H′), with KSA values adopted from Bärring et al. (2002). Concerning the way of deriving the KSA it would be important to provide estimates of uncertainties that would be introduced by this assumption. Moreover, it is unclear how this adopting was performed for B(a)P since the publication of Bärring et al. (2002) was focused on experimental determination of the soot–water distribution coefficients for PCDDs, PCDFs, and PBDEs.

Page 6, line 16: There would be a need to describe more clearly these two parameterizations for KOA (e.g. to give equations, to show difference).

Page 10, line 6: It would be better to provide the difference (relative or absolute) between the maps in Figures 2a and 2b instead of direct comparison.

Page 21, Table 1: it is not shown from where the values for Ksoot-water, Ksoot-air, and OH reaction rate were taken.

Page 22, table 2: Table 2 does not correspond to its description in section 3.2 for 4th and 5th scenarios.

Page 27, figure 3: Please, correct the abbreviations JB-W and DL, as they were not defined earlier. The same for figure S5. Concerning the maps, it would better to show the difference between them in relative units.

Table 4 and Table S8 do not present mean modeled and observed B(a)P concentrations. In the column 'Model' both abbreviations for scenarios and for GPP models are used (for example, JP-W and DE, HB, etc.). Please, provide equations or references for IOA etc.

---

## Author Comment (AC1) · 18 May 2016

This manuscript presents an evaluation of alternative gas-particle partitioning schemes in a modified version of the WRF-CMAQ model run for Benzo-a-pyrene (BaP). The authors compare modeled BaP concentrations within a European model domain under 5 different gas-particle partitioning schemes and also compare model results to measurement data from EMEP monitoring sites.

The novel aspect of this work seems to be announcing that the different gas-particle partitioning schemes have been incorporated into CMAQ. The more complicated schemes apparently provide the best model performance relative to measurement data, but the authors state that the disagreements between measurements and the model are likely mostly attributable to uncertainties in emission estimates. Therefore it is difficult to draw conclusions about which partitioning scheme is best based just on results reported in this paper.

We appreciate the thoughtful review and the recognition of the novelty of the approach in 3D air quality systems such as CMAQ, despite the limited spatiotemporal coverage of the EMEP network.

I have two scientific concerns about the paper. I think both are significant, but possibly #1 arises from unclear presentation in the paper.

1) I understand Table 2 and the text to say that Scenarios 3, 4 and 5 include both the Junge-Pankow adsorption model *and* the Harner-Bidleman Koa absorption model. Is that correct? If so, I think the model is perhaps adding two redundant descriptions of gas-particle partitioning.

It has been a while since I looked at the JP model, but my recollection is that it includes a fitting parameter that was derived from empirical data under the assumption that all partitioning was by adsorption to aerosol surfaces. But, the assumption that adsorption dominates could not be confirmed. Xiao and Wania (http://www.sciencedirect.com/science/article/pii/S1352231003002139) later showed that vapor pressure and Koa both describe sorption to organic matter equally well. Therefore, I think partitioning schemes that include both vapor pressure and Koa as chemical parameters are likely including redundant information, and could easily be over-fitted.

Reply:

We agree, the parameter $c_J$ (high for strong sorption, low for weak sorption) in the JP model may not be totally free from absorptive contributions. Therefore, adding $\theta$ of JP and $K_{oa}$ may eventually lead to overpredictions, but these should be very low (negligible), as the absorptive contribution within JP is very low. It is very low, as the fitting was done for all possible vapors, and many of these were inorganic (almost not soluble in octanol). Unfortunately, no $c_J$ for BAP based on measurements has ever been published. However, we found recently in background aerosol of Central Europe (Kosetice 2012-13, n=162 aerosol samples, $c_J$ derived from experimental S/V data; Shahpoury et al., 2016) that $c_J$ for BaP is very low, namely 0.01-5.46 Pa cm (median = 0.44 Pa cm; and also low for all other PAHs, with one exception, anthracene for which $c_J > 17.2$ Pa cm is found). This finding is not representative for all types of aerosol in the simulation, but for the European background aerosol. It supports the perception that a possible implicit absorptive contribution to PAHs' $c_J$ must be very low i.e., negligible. Therefore, the combined partitioning according to both the JP and a $K_{oa}$ model is justified (note that the Harner-Bidleman version of the $K_{oa}$ model is present only in scenario 3 – Table was modified to avoid confusion).

New text in the revised version reads (after line 13, page 5): "The empiric JP model, despite assuming adsorption to be the only relevant partitioning process, may eventually not be free from absorptive contributions (to $c_J$), but these should be very low. The fitting in the empiric parameterisation was done for all possible vapors, many of these not soluble in octanol. No $c_J$ for BAP derived from ambient measurements has ever been published. However, we found recently in background aerosol of Central Europe (Kosetice 2012-13, n=162 aerosol samples, $c_J$ derived from experimental S/V data; Shahpoury et al., 2016) that $c_J$ for BAP is very low, namely 0.01-5.46 Pa cm (median 0.44 Pa cm), supporting the perception that a possible implicit absorptive contribution to $c_J$ must be very low."

Shahpoury P., Lammel G., Albinet A., Sofuoglu A., Domanoğlu Y., Sofuoglu C.S., Wagner Z., Zdimal V.: Evaluation of a conceptual model for gas-particle partitioning of polycyclic aromatic hydrocarbons using poly-parameter linear free energy relationships, Environ. Sci. Technol., Submitted, 2016.

2) The authors state that soot-air partition coefficients were calculated as the ratio of soot-water adsorption constants (Ksw) and the Henry's Law constant. Kai Goss has pointed out that applying a thermodynamic triangle for a solid interface in contact with water and the same solid interface in contact with an adjacent gaseous phase is a conceptual mistake that can lead to very large errors (http://pubs.acs.org/doi/abs/10.1021/es0301370).
I actually do not see a good option for the authors to overcome this problem, since the Lohmann & Lammel (2004) and the Dachs & Eisenreich (2000) papers that are the basis for the gas-particle partitioning scheme applied in this paper are both based on thermodynamic triangles that Goss argues (convincingly!) are invalid.
Reply:
We thank the reviewer for pointing this out. We agree that the method applied is imperfect, but it is the best available and, hence, commonly accepted in atmospheric modelling (as cited by the referee, Galarneau et al., 2014, and others). The only other option available (suggested by Dachs et al., 2004, and van Noort, 2003) parameterises $K_{soot-air}$ as a function of the BC specific surface area, which is basically unknown and certainly transient while aging (typically changing by at least one order of magnitude within one day following emission from combustion sources). Dachs concludes that prediction of $K_{sa}$ by either method provide reasonably similar results.
New text in the revised version reads (after line 24, page 6): "This method is subject to uncertainties (Goss, 2004), but is accepted and suitable (Dachs et al., 2004; besides others)."

van Noort, P.: A thermodynamics-based estimation model for adsorption of organic compounds by carbonaceous materials in environmental sorbents, Environ. Toxicol. Chem., 22(6), 1179–1188, doi:10.1002/etc.5620220601, 2003.
Dachs, J., Ribes, S., van Drooge, B., Grimalt, J., Eisenreich, S. J. and Gustafsson, Ö.: Response to the comment on "Influence of soot carbon on the soil−air partitioning of polycyclic aromatic hydrocarbons," Environ. Sci. Technol., 38(5), 1624–1625, doi:10.1021/es0307118, 2004.
Goss, K.-U.: Comment on "Influence of soot carbon on the soil−air partitioning of polycyclic aromatic hydrocarbon," Environ. Sci. Technol., 38(5), 1622–1623, doi:10.1021/es0301370, 2004.

---

## Author Comment (AC2) · 18 May 2016

General comments

The manuscript describes the application of community air quality model CMAQ to study atmospheric transport of B(a)P and effects of gas-particle partitioning and degra- dation. Several gas-particle partitioning models were implemented in the CMAQ model extending its capabilities with respect to modeling of PAHs. Though the subject of the study is of importance with regard to existing air quality problems in Europe, description of this study suffers from obscurity and thus requires substantial revisions.

First of all, I would mention not clear enough description of parameters applied in the equations 10 and 11, related to gas-particle partitioning (e.g. KOA, KSA). Secondly, evaluation of the effects of several GPP mechanisms through their incremental testing (incremental testing is defined in section 3.2 and Table 2) is not quite clearly described with regard to sensitivity of CMAQ model output to particular partitioning mechanisms in model simulation scenarios. In some cases abbreviations, defined for GPP models and scenarios and used throughout the analysis, are mixed like e.g. DE model and DE scenario. This leads to problems with understanding what is presented, for example, in the second column of tables 4 and S8 called 'Model'. There are both abbreviations for scenarios and models (for example, JP-W and DE, HB, etc.). Finally, the underestimation of observed B(a)P concentrations in air is explained by low emissions. At the same time, there is no discussion of potential problems of emission data used, no comparison of total annual emissions with other studies, for example, with similar studies e.g. Aulinger et al. (2007) and others.

Reply:
We appreciate the very thorough and detailed review along with the recognition of the potential relevance of the presented study. The first two points of concern have been addressed in a revised manuscript. Regarding the last area of concern, we feel that the manuscript adequately discusses the limitations of emission data used, which have been the major topic of work published by the authors (Bieser et al., 2012).

Specific comments

Page 3, line 12-13: "CMAQ contains modules representing advection, eddy diffusion, in-cloud, and precipitation processes". It would be better to use 'in cloud and below- cloud scavenging with precipitation'.

Reply: Corrected

Page 3, line 26: 'gas phase reactions' instead of 'gas reactions'.

Reply: Corrected

Page 4, line 23: Though the equation 2 follows the publication of Aulinger et al. (2007), it is not clear how the particulate fractions in each mode i, fi, are obtained, because in the cited work (Cooter and Hutzell, 2002) similar equation is written for the sum of partition coefficients, but not for fractions of a compound in particulate phase.

Reply: According to equations 1-4 in Cooter and Hutzell (2002), particulate fractions are defined for each mode i of CMAQ aerosol.

Page 5, equation 7: fOM is used without index i. Does it mean that the fraction of organic matter in aerosol particles is the same in each of three modes?

Reply: Corrected throughout the text

Page 6, line 3: assumptions used here for ratio of activity coefficients and ratio of mean molar weights of organic matter of the particles and octanol (that they are equal unity) need to be discussed with regard to their uncertainties as it was shown in e.g. (doi: 10.5194/acpd-14-21341-2014 ).

Reply: More explanations will be added. The new text reads (after line 7, page 6): "Assuming that octanol imitates organic matter in PM, Harner and Bidleman (1998) suggested that the ratio of $\gamma_{oct}/\gamma_{OM}$ and $M_{oct}/M_{OM}$ can be assumed to be 1. However, it was later suggested that $M_{OM}$ could be much higher, particularly in secondary organic aerosols containing polymeric structures (Kalberer et al., 2004); a mean value of 500 g mol$^{-1}$ was later suggested by Götz et al. (2007), which results in $M_{oct}/M_{OM}$ of 0.26."

Götz, C. W., Scheringer, M., MacLeod, M., Roth, C. M. and Hungerbühler, K.: Alternative approaches for modeling gas−particle partitioning of semivolatile organic chemicals: model development and comparison, Environ. Sci. Technol., 41(4), 1272–1278, doi:10.1021/es060583y, 2007.
Kalberer, M., Paulsen, D., Sax, M., Steinbacher, M., Dommen, J., Prevot, A. S. H., Fisseha, R., Weingartner, E., Frankevich, V., Zenobi, R. and Baltensperger, U.: Identification of polymers as major components of atmospheric organic aerosols, Science, 303(5664), 1659–1662, doi:10.1126/science.1092185, 2004.

Page 6, line 14-15: Soot-air partition coefficients were calculated as the ratio of soot water adsorption constants KSW and the inverse Henry's Law constant (HC), with KSA values adopted from Bärring et al. (2002). Concerning the way of deriving the KSA it would be important to provide estimates of uncertainties that would be introduced by this assumption. Moreover, it is unclear how this adopting was performed for B(a)P since the publication of Bärring et al. (2002) was focused on experimental determination of the soot−water distribution coefficients for PCDDs, PCDFs, and PBDEs.

Reply: $K_{sa}$ was calculated based on $K_{sw}$ (typo fixed) following the approach described in the footnotes of Table 1 in Lohmann and Lammel (2004). Despite the publication title not mentioning PAHs, this value is also referenced in Table 4 of Bärring et al. (2002). This information will be included in Table 1 for clarity. We agree that the method applied is imperfect, but it is the best available and, hence,

commonly accepted in atmospheric modelling (as cited by the referee, Galarneau et al., 2014, and others). The only other option available (suggested by Dachs et al., 2004, and van Noort, 2003) parameterises $K_{soot-air}$ as a function of the BC specific surface area, which is basically unknown and certainly transient while aging (typically changing by at least one order of magnitude within one day following emission from combustion sources). Dachs concludes that prediction of $K_{sa}$ by either method provide reasonably similar results.

New text in the revised version (following a suggestion of the other reviewer) reads (after line 24, page 6):
"This method is subject to uncertainties (Goss, 2004), but is accepted and suitable (Dachs et al., 2004; besides others)."

van Noort, P.: A thermodynamics-based estimation model for adsorption of organic compounds by carbonaceous materials in environmental sorbents, Environ. Toxicol. Chem., 22(6), 1179–1188, doi:10.1002/etc.5620220601, 2003.
Dachs, J., Ribes, S., van Drooge, B., Grimalt, J., Eisenreich, S. J. and Gustafsson, Ö.: Response to the comment on "Influence of soot carbon on the soil−air partitioning of polycyclic aromatic hydrocarbons," Environ. Sci. Technol., 38(5), 1624–1625, doi:10.1021/es0307118, 2004.
Goss, K.-U.: Comment on "Influence of soot carbon on the soil−air partitioning of polycyclic aromatic hydrocarbon," Environ. Sci. Technol., 38(5), 1622–1623, doi:10.1021/es0301370, 2004.

Page 6, line 16: There would be a need to describe more clearly these two parameterizations for $K_{oa}$ (e.g. to give equations, to show difference).

Reply: The two $K_{oa}$ parameterisations are described in the cited publications. Previous modelling studies of B(a)P followed either Beyer et al. (Aulinger, 2007) or Odabasi et al. (Galarneau, 2014). The relevant text was amended to explicitly state that Odabasi et al. determine $K_{oa}$ as a function of temperature.

Page 10, line 6: It would be better to provide the difference (relative or absolute) between the maps in Figures 2a and 2b instead of direct comparison.

Reply: The goal of Figure 2 is to provide an illustation of B(a)P distributions and gradients across Europe. For better clarity we selected logarithmic scale. The difference is in fact provided in Figure 3 that provides the absolute difference comparison plot, discussed in the following paragraph.

Page 21, Table 1: it is not shown from where the values for $K_{soot-water}$, $K_{soot-air}$, and OH reaction rate were taken.

Reply: Corrected

Page 22, table 2: Table 2 does not correspond to its description in section 3.2 for 4th and 5th scenarios.

Reply: Corrected

Page 27, figure 3: Please, correct the abbreviations JB-W and DL, as they were not defined earlier. The same for figure S5. Concerning the maps, it would better to show the difference between them in relative units.

Reply: Corrected the abbreviations uniformly throughout the text. However, no visual benefit when moving from absolute to relative units was noticeable in the figures, therefore omitted.

Table 4 and Table S8 do not present mean modeled and observed B(a)P concentrations. In the column 'Model' both abbreviations for scenarios and for GPP models are used (for example, JP-W and DE, HB, etc.). Please, provide equations or references for IOA etc.

Reply: Corrected the abbreviations throughout the text and according to previous comment. The metrics of performance have been calculated based on the openair R package (Carslaw and Ropkins, 2012) as noted in section 3.2 (equations/references can be found therein).

---

## Author Response (AR2)

**Response to co-Editor**

Comments to the Author:
Thank you for the revised version of the manuscript. While some effort has been made to address the original reviewer comments, unfortunately the most critical concerns have not been satisfactorily addressed in the revised manuscript. Specifically, I am referring to 1) the redundancy of the partitioning schemes (raised by original Referee #1) and 2) the validity of the scheme by Lohmann and Lammel (2004) which has been challenged in the literature, and has been raised by all three reviewers. Furthermore, the new Referee #3 also raises concerns about the novelty of the work as compared with what has already been done.

For a revised manuscript to be reconsidered, the authors need to do a better job in convincingly arguing for and describing the approaches they have used and the novelty of their work. Regarding the first point, indeed it seems redundant to use both saturation vapor pressures and partitioning coefficients in an additive manner as pointed out by Referee #1. If the authors disagree, convincing arguments on the grounds should be provided. Regarding the second point, it is not enough to state that an approach is "accepted and suitable" but instead provide a thorough argumentation on why the arguments made by Goss (Environ. Sci. Technol. 2004) are invalid or can be neglected, along with (if indeed the approach is best available) a thorough discussion on the uncertainties associated with the chosen approach as suggested by the original Referee #2. Furthermore, the novelty of the work needs to be clearly justified and the previous work on the topic needs to be adequately discussed (see comments by Referee #3).

Besides these critical concerns, the new Referee #3 also provides some useful suggestions on potential improvements on the paper, which I recommend you consider if you wish to submit a revised manuscript, along with a ensuring the presentation of the manuscript to be as clear and transparent as possible.

If you think that you can sufficiently address these concerns, a revised manuscript can be reconsidered, but a further review will be necessary. Due to the significant time this would likely require, I also fully understand if you choose to submit the manuscript elsewhere.

I am sorry I cannot be more positive at this point,

best regards,

Ilona Riipinen

We would like to thank the Co-Editor for the valuable remarks and suggestions to further clarify the concerns raised by the referees. In order to satisfactory address those points, a revised manuscript has been submitted. With respect to the first point, a more elaborate justification is presented as a response to Referee #1. Regarding the second point, we would like bring to the co-editor's attention that the validity of the Lohmann and Lammel scheme (Environ. Sci. Technol. 2004) has not been raised by all

three reviewers. Moreover, neither the Dachs-Eisenreich scheme nor the Lohmann and Lammel scheme have ever been criticized (instead the scheme suggested by Lohmann & Lammel represents the state of the art for PAH partitioning, has been widely used, and the paper has been cited some 170 times). What has been criticized is, the derivation of one of the parameters used in the scheme, $K_{SA}$, i.e., namely its derivation from $K_{ow}$ (Goss, Environ. Sci. Technol. 2004) as done in the originally submitted version. This criticism was not published after Lohmann and Lammel (2004), but at the same time. As has been pointed out (in our subsequent reply to the first reviews), an alternative exists ($K_{SA}$ derived as a function of the BC specific surface area; suggested by Dachs et al., Environ. Sci. Technol. 2004, and van Noort, Environ. Toxicol. Chem. 2003), which is not subject to a principle weakness. We have included this additional parameterisation in the revised version. With respect to the novelty of this work, we feel it adequately serves as a foundation model that currently undergoes further development with respect to PAH/POP modelling (i.e. further species for which aerosol processes and cycling between air and ground compartments are essential). Moreover, as we note in our comments to Referee #3, the very specific novelty and focus of this work (i.e first study that evaluates a wide range of gas/particle partitioning schemes in CMAQ) have been pointed out by Referee #1 and #2, as well as in the most recent publication in the field by Galarneau et al. (Atmos. Chem. Phys., 2014, p. 4075), where the relevance of a similarly titled manuscript was highlighted (Galarneau, E.: Evaluation of particle/gas partitioning in a regional air quality model (AURAMS-PAH), in preparation, 2014).

Despite the time and (computational) resource constraints, we have performed additional model simulations and the results are included in the revised version.

**Anonymous Referee #1**

After reviewing the author's response to earlier comments from myself and another reviewer, I recommend rejecting the paper for final publication.

The novel contribution of the paper is the incorporation of gas/particle partitioning schemes into CMAQ. However, the authors have made a mess of the job by 1) implementing redundant schemes (the JP and Koa models implemented additively) and 2) a scheme that has been convincingly argued in the literature to be conceptually flawed (Lohmann & Lammel 2004).

These two criticisms are basically the same as I offered in my first review of the paper. The arguments and small changes to the paper the authors offered in response are not satisfying to me.

With regard to point 1: The sub-cooled liquid vapor pressure of the PAHs is essentially directly proportional to their Koa. Thus there is no mathematical difference between schemes based on vapor pressure and Koa except constants in the equations and the conceptual interpretation of the meaning of those constants. Adding these two schemes together does not provide insight about the relative contributions of adsorption versus absorption processes for the PAHs. The author's arguments about other chemicals used in fitting the JP relationship do not seem relevant to me.

Reply: We disagree, as:

(a) Proportionality of sub-cooled liquid vapor pressure and $K_{oa}$ can mean, but does not necessarily mean that each pair of single-parameter linear free energy relationships (spLFERs), one using $p_L$, the other $K_{oa}$ are 'mathematically redundant'. This depends on the data sets regressions have been fitted on.

(b) Proportionality of sub-cooled liquid vapor pressure and $K_{oa}$ is expected for substances which gas-particle partitioning is determined by absorption into OM (Pankow, Atmos. Environ. 1998; Xiao and Wania, Atmos. Environ. 2003). Accordingly, for substance classes which gas-particle partitioning is in fact dominated by absorption into OM i.e., chlorobenzenes, PCBs, PCNs, PCDD/Fs, PBDEs (Finizio et al., Atmos. Environ. 1997; Harner and Bidleman, Environ. Sci. Technol. 1998, besides others), Xiao and Wania (2003) find the descriptors $p_L$ and $K_{oa}$ 'very highly correlated' (see the abstract), not for PAHs. Proportionality of sub-cooled liquid vapor pressure and $K_{oa}$ is furthermore expected for substances which gas-particle partitioning is less determined by absorption into OM, because of the uncertainties of the experimental data (related to sampling artefacts biasing the observed phase distribution, to temperature variation during sample collection), the uncertainties of the substance properties (quality and consistency e.g., experimentally inaccessible sub-cooled liquid vapor pressure) (as both pointed out by Xiao and Wania, 2003), and the temperature sensitivity of other types of molecular interaction of these substances with PM than sorption into OM. In other words, a correlation for apolar molecules such as PAHs is not a surprise, but does not necessarily indicate that their gas-particle partitioning is determined by absorption into OM.

(c) For PAHs it has been shown (Lohmann and Lammel, 2004) that assuming solely absorption to explain gas-particle partitioning ($K_{oa}$ model e.g., Finizio et al., 1997) is deficient, while a dual model which considers also adsorption to soot (Dachs and Eisenreich, 2000) is superior i.e., better predicting observed partitioning at a large variety of sites. This parameterisation (Lohmann and Lammel, 2004) has been representing the state of the art, which is reflected in numerous (>> 50) applications for PAH gas-particle partitioning in the literature, including applications testing various models (Lammel et al., Chemosphere 2009; Wang et al., Atmos. Environ. 2013; Wang et al., Environ. Sci. Pollut. Res. 2013; Friedman et al., Environ. Sci. Technol. 2014; Galarneau et al., Atmos. Chem. Phys. 2014; Sangiorgi et al., Environ. Sci. Pollut. Res 2014; Liu et al., Chemosphere 2015; Li et al., Atmos. Environ. 2016; besides others).

(d) Adsorptive contributions to the overall sorption process (such as being described by the Junge-Pankow model) and absorptive contributions (such as into a liquid or semi-solid organic phase being described by the so-called $K_{oa}$ model) are indeed expected to behave almost additive, as these single parameter linear free energy relationships go with the assumption that the one parameter describes the dominant of all possible types of molecular interactions of an organic molecule with the matrix (see e.g. Schwarzenbach et al., Env. Organic Chem. 2$^{nd}$ ed, Wiley 2003, ch. 11.2; Goss and Schwarzenbach, Environ. Sci. Technol. 2001, 35, 1-9), while other interactions are negligible and not captured. There is a small 'redundancy' when combining the Junge-Pankow with a $K_{oa}$ model (scenarios 3-5), which has been detailed in the reply to the first round of reviews, repeated here: We agree, the parameter $c_J$ (high for strong sorption, low for weak sorption) in the Junge-Pankow (JP) model may not be totally free from absorptive contributions. Therefore, adding $\theta$ of JP and $K_{oa}$ may eventually lead to overpredictions, but these should be very low (negligible), as the absorptive contribution within JP is very low. It is very low, as the fitting was done for all possible vapors, and many of these were

inorganic and polar (almost not soluble in octanol). Unfortunately, no $c_J$ for BAP based on measurements has ever been published. However, we found recently in background aerosol of Central Europe (Kosetice 2012-13, n=162 aerosol samples, $c_J$ derived from experimental S/V data; Shahpoury et al., to be published in Environ. Sci. Technol. 2016, following minor revision) that $c_J$ for BaP is very low, namely 0.01-5.46 Pa cm (median = 0.44 Pa cm; and also low for all other PAHs, with one exception, anthracene for which $c_J$>17.2 Pa cm is found). This finding is not representative for all types of aerosol in the simulation, but for the European background aerosol. It supports the perception that a possible implicit absorptive contribution to PAHs' $c_J$ must be very low i.e., negligible. Therefore, the combined partitioning according to both the JP and a $K_{oa}$ model is justified (note that the Harner-Bidleman version of the $K_{oa}$ model is present only in scenario 3 – Table was modified to avoid confusion).

Finally, we should also stress that so far in literature, all regional-scale air pollution modelling studies that utilize the aerosol module AERO of CMAQ and employ both adsorptive (JP scheme) and absorptive contributions ($K_{oa}$ variants) to the overall sorption process, did so in an additive manner (see Table S1 – i.e Aulinger et al., 2007, Matthias et al., 2009, Bieser et al., 2012, Silibello et al., 2012). When bringing empirical models into a mechanistic modelling framework it is important to understand both the limitations of the empirical schemes (i.e. dependency on the data sets regressions have been fitted on, addressed in previous paragraphs), as well as the limitations posed by the predefined modelling framework where they are brought to interact (i.e., CMAQ's aerosol module AERO). While CMAQ has one of the most advanced aerosol modules among regional air quality models, certain simplifications in the treatment of coarse PM and OM (e.g., CMAQ does not include organic materials in coarse PM; however measurements show that organics constitute a significant component, see Binkowski, J. Geophys. Res. 2003; Li et al., Atmos. Chem. Phys., 2013) are posing limitations to fully exploit the advanced empirical gas-particle partitioning models ($K_{oa}$, Dachs-Eisenreich) in this specific mode. Furthermore, the large discrepancies in the OM and BC emission inventories (i.e. 2000-2001 SMOKE-EU totals were found to be a factor of 5 lower over Europe compared to previous estimates for the year 1995; Kupiainen and Klimont, Atmos. Environ. 2007; Matthias et al., Atmos. Chem. Phys. 2008) along with limited availability of observations have a considerable impact on the performance of these advanced schemes. An implicit overlap of the JP model with the $K_{oa}$ (or another absorption) model causes a small inconsistency, much smaller than existing uncertainties of output aerosol parameters propagating from emission estimates and from the current design and state of development of the CMAQ aerosol module (AERO). Although the importance of organic aerosol has been noted in the CMAQ literature (Mathur et al., J. Geophys. Res. 2008; Matthias et al., Atmos. Chem. Phys. 2008), measurements in Europe are still sparse and a detailed assessment could not be done in this study. A future more sophisticated and consistent CMAQ aerosol module will expectedly allow to identify and better quantify remaining inconsistencies of gas-particle partitioning.

New text in the revised section of the conclusions (after line 8, page 14) reads: "In addition, certain simplifications in the treatment of coarse PM and OM within the aerosol module of CMAQ (Binkowski, 2003; Mathur et al., 2008) are posing limitations to fully exploit the advanced empirical gas-particle partitioning models ($K_{oa}$, Dachs-Eisenreich) in this specific mode. An implicit overlap of the empiric adsorption model (JP) with the $K_{oa}$ (or another absorption) model (see section 2.1.1) may cause a small

inconsistency, which is considered to be negligible compared to existing uncertainties of output aerosol parameters propagating from emission estimates and from the current design and state of development of the CMAQ aerosol module (AERO)."

With regard to point 2: I do not agree that a model that is "commonly accepted" but that has been shown to be conceptually flawed should continue to be applied and propagated in the literature.

Reply: We disagree. This is probably a misunderstanding by the reviewer: The Dachs-Eisenreich scheme or Lohmann and Lammel scheme (have to our) knowledge never been criticized in the literature. Their common acceptance is documented by >>50 applications in the literature including a number of comparative usage against other schemes (Lammel et al., Chemosphere 2009; Wang et al. Atmos Environ 2013; Wang et al. Environ Sci Pollut Res 2013; Friedman et al Environ Sci Technol 2014; Galarneau et al., Atmos Chem Phys 2014; Sangiorgi et al., Environ Sci Pollut Res 2014; Liu et al., Chemosphere 2015; Li et al., Atmos Environ 2016; besides others). What has in fact been criticized is the derivation of one of the parameters used in the scheme, $K_{SA}$, i.e., namely its derivation from $K_{ow}$ (Goss Env Sci Technol 2004), as was done in the originally submitted version (see also reply to comments of the editor, above). However, the results achieved with this conceptually flawed method, were found to be similarly good, than those derived using the alternative method (Dachs et al. Environ Sci Technol 2004, 38, pp 1624–1625). In the revised version we use another, not conceptually flawed method to derive $K_{soot/air}$ (see above).

**Anonymous Referee #3**

This paper describes the application of the CMAQ model to simulate benoz[a]pyrene (BaP), one of the more toxic polycyclic aromatic hydrocarbons (PAHs). Several different parameterizations of gas-particle partitioning are implemented in the existing aerosol module and evaluated against one another. While this study represents a contribution to scientific progress with the new capability for modeling BaP within CMAQ, much of what is presented in terms of the gas-particle partitioning scheme comparisons has already been shown in other modeling studies, both at regional and global scales. The paper is also lacking detail, both in the description of the model construction and specifics about the results shown, but also in the form of little discussion of previous work and how these results compare. I recommend a resubmission after major revision. Specifically, my concerns are:

The novelty is the implementation of BaP behavior in CMAQ, but not to a regional model (first paragraph, page 3; Galarneau et al., ACP 2014, already cited within, compared these schemes within a regional model).

We would like to thank the referee for the constructive comments. In fact, BaP had first been modelled using CMAQ by Aulinger et al. (2007). Instead, the novelty of our study lies precisely in a more extensive evaluation of gas-particle partitioning schemes currently available for BaP within an air

quality model such as CMAQ. Furthermore, Galarneau et al., (ACPD 2014) as erroneously mentioned, have not compared the full spectrum of the GPP schemes presented here (less than half), nor did all other studies (see Table S1). The novelty of the present study was indeed inspired by the research gap pointed in Galarneau et al., 2014, where the relevance of a similarly titled manuscript was highlighted (Galarneau, E.: Evaluation of particle/gas partitioning in a regional air quality model (AURAMS-PAH), in preparation, 2014; - not yet published, to our best knowledge).

The authors characterize the work as an extension of CMAQ to POPs, but "POPs" covers a very large set of chemicals and the model is tested for only one compound. I recommend at a minimum that "PAHs" is substituted for "POPs", even though only one PAH is tested.

Reply: POPs indeed cover a large number of chemicals with varying properties, however the rest of the sentence makes it clear that the focus is on "an adaptable framework that accounts for gaseous chemistry, heterogeneous reactions, and gas-particle partitioning (GPP)". The modelling framework and routines implemented within CMAQ are largely the same, and only these property-specific algorithms need to be (and have been) modified for other POPs. In addition, the introductory paragraphs explain and reference that some PAHs - and its most well studied representative (BaP) - are considered POPs.

There is at least one mischaracterization of previous work within the literature review and I recommend the authors check for others: Friedman and Selin 2012 (line 28, page 2) presented a meteorologically driven PAH model, not a general circulation model.

Reply: The work by Friedman and Selin, 2012, is based on GEOS-Chem, a global 3-D chemical transport model which has also been adapted to run with climate data from the NASA/GISS general circulation model. In order to take into account the suggestion, the term general circulation model was replaced by chemical transport model for clarity.

Recent work has suggested that the equilibrium gas-particle partitioning schemes tested here, which were first presented over a decade ago (Lohmann and Lammel, 2004), may not be representative of partitioning during long-range transport (see DOI: 10.1021/es302743z and DOI: 10.1021/es405219r). There is no discussion related to the hypothesis that SOA may facilitate PAH transport and I think there needs to be (even if brief).

Reply: We agree, SOA play an important role in long-range transport of POPs and a brief mention of the so-called "burial effect" was already mentioned in section 2.2. Additional discussion is provided at the end of the manuscript that now reads (after line 25, page 14): "Following modelling studies should focus on quantifying the long-range transport potential and examining the hypothesis that secondary organic aerosol (SOA) may facilitate PAH transport by utilising more sophisticated aerosol and heterogeneous chemistry parameterisations or submodels (i.e. accounting also for the burial effect - cf. Sect. 2.2). More specifically, recent evidence suggests that in cold and dry air accessibility of PAHs in OM is reduced (due to low diffusivity), which might explain the apparent inconsistency of high LRT potential (BaP levels in the Arctic) on one hand side and relative high heterogeneous reactivity

measured in the laboratory (Friedman et al., 2014; Zelenyuk et al., 2012; Zhou et al., 2012; 2013). As understanding of PAHs' atmospheric lifetimes, PAHs' interaction with SOA, and chemical composition of ambient OM progresses, a new need for additional studies quantifying GPP and LRT potential under a wide range of atmospheric conditions is emerging. In view of such findings, the WRF-CMAQ-BaP modelling system should be extended to study a wide range of additional organic pollutants and processes (i.e. multicompartmental cycling, biodegradation, heterogeneous chemistry)."

The manuscript is quite brief and lacks context. Is this model an improvement over other European PAH models? I found no comparison to results from other modeling studies from the same region, and no justification or motivation for choosing the particular model domain. There also could be a much greater discussion of how these results, particularly for the gas-particle partitioning schemes and the heterogeneous O3 reactions, compare to results of other modeling studies; there is indeed a number of modeling studies already published that examine the same partitioning schemes and employ the same O3 reaction rates.

Reply: The goal of this study was not to provide with a comparison against previous work on the field, but rather with a more complete approach in treating gas-particle partitioning. Even if some of the studies performed over Europe using a regional air quality model, it was with different model or aerosol module, GPP schemes, and emissions. The domain is within the area covered by the UNECE-CLRTAP (and its POP protocol, covering PAHs) and was specifically chosen such as to exploit recently prepared high-resolution emission data of one of the authors (Bieser et al., Geosci Model Dev 2012) and make use of the only monitoring network in the region (EMEP). BaP heterogeneous chemistry, even with ozone, let alone other oxidants, is incompletely studied (few model aerosol materials, limited temperature and humidity ranges, phase states and particle morphologies). Therefore, the test of a number of scenarios of reactivity, in line with present lab and field knowledge would be a study of its own. Such modelling is under way, but not subject of the present study. Of course, a model performance comparison would be of great value and could be pursued in the future and could be feasible under the auspices of a larger intercomparison study such as AQMEII (http://aqmeii-eu.wikidot.com/).
New text in the revised version reads (after line 27, page 8) : "The domain is within the area covered by the UNECE-CLRTAP (and its POP protocol, covering PAHs) and was specifically chosen such as to exploit recently prepared high-resolution emission data (Bieser et al., 2011) and make use of the only monitoring network in the region (EMEP, see below)"

There is no quantitative discussion of the uncertainties contributed by physicochemical properties or emissions.

Reply: Regarding the quantitative discussion of the uncertainties introduced by the BaP emission inventory, this has been the direct topic of a study that has already been published by one of the co-authors (as pointed out previously to Referee #2). With respect to the physicochemical properties, an additional scenario and simulation was performed to investigate uncertainties contributed by different methodologies to calculate $K_{oa}$. In addition, the revised version of the manuscript now includes another

simulation which has been performed to address uncertainties contributed by different methods in calculating $K_{SA}$.

New text in the revised version reads (after line 27, page 6): "As a step to address uncertainties contributed by different methods in calculating $K_{SA}$, a thermodynamic estimation model suggested by van Noort (2003) was also tested:

$$\log K_{SA} = -0.85 \log p_L^\circ + 8.94 - \log(998/a_{soot}) \qquad (12)$$

The soot specific surface area was set to 18.21 $m^2$ $g^{-1}$, derived as the geometric mean of surface areas for traffic, wood, coal, and diesel soot (i.e. 59.4, 3.6, 8.2, and 62.7 $m^2$ $g^{-1}$, respectively) (Jonker and Koelmans, 2002). Additionally, in order to estimate uncertainties arising from $K_{oa}$ and its temperature dependence, two different parameterisations were tested, the first based on the work of Beyer et al. (2002) and the second following the temperature-dependent parameterisation suggested by Odabasi et al. (2006)."

(after line 23, page 12): "In addition, simulations that address the uncertainties contributed by the method of estimating $K_{SA}$ were performed for the winter period of 2006 using the DE scheme. Figure S6 reveals similar results with BaP levels slightly higher in cells with strong emissions sources. However, with the exception of Moscow, the average difference BaP concentration in urban cells did not exceed 0.02 ng $m^3$ when $K_{SA}$ was calculated based on the thermodynamic estimation model suggested by van Noort (2003). Similarly to $K_{oa,}$ results suggest that the partitioning based on each $K_{SA}$ approach is sufficiently similar."

Table 2 is unclear. It seems to imply that a given scenario can include both the Junge-Pankow scheme and the Harner-Bidleman or the Dachs-Eisenreich scheme. This would be redundant; they are different approaches for modeling the same phenomenon.

Reply: minor/negligible 'redundance', see response to the Referee #1 above;

Something that would significantly enhance the contribution of this study to the current science is to apply the presented gas-particle partitioning schemes (once fully evaluated against measurements) to air-surface exchange (it is unclear if surface-air exchange is considered in the model presently). The authors mention that a) air-surface exchange is an area of POPs transport with considerable uncertainty and b) their framework could be extended to simulate mass exchange between different environmental compartments – to do so here would demonstrate the utility of the POPs modeling capability in CMAQ and add more novelty to the manuscript.

Reply: We agree, but an air-surface model is of little relevance when applied to BaP. As this is an extreme task to undertake in this very same publication (new species, compartments, air-surface exchange module with inputs and parameterisations), we are reserved to present and evaluate it in a subsequent manuscript once the foundation model in its current form has been fully evaluated.

The definition of an abbreviation is not always given on first use (e.g., WRF).

Reply: Definition of the abbreviation for WRF was misplaced later in the text. Corrected for the first instance.

It is not clear, but it seems that all of the concentration comparisons are conducted for total (gaseous and particulate) BaP concentrations. There is no comparison of simulated gas-particle speciation to observations of fraction in the particle versus gas phase. This data is available for Europe and should be included to fully evaluate the accuracy of the different schemes (a comparison between schemes has little use without any ground-truthing).

Reply: This is because there simply is no data available for the gas phase BaP for Europe (the striking majority of observations < LOQ, EMEP reports total gas+aerosol, no other data available). BaP is expected to rapidly move from gas to particulate phase close to combustion sources, or even be mostly emitted in particulate phase. This has been also confirmed by preliminary calculations of particulate mass fraction, $\theta$, for the simulations presented in this work, which indeed show very low gas-phase BaP (even in tests treating it as non-reactive gas, similar to Aulinger et al., 2007).

Anonymous Referee #1

This manuscript presents an evaluation of alternative gas-particle partitioning schemes in a modified version of the WRF-CMAQ model run for Benzo-a-pyrene (BaP). The authors compare modeled BaP concentrations within a European model domain under 5 different gas-particle partitioning schemes and also compare model results to measurement data from EMEP monitoring sites.
The novel aspect of this work seems to be announcing that the different gas-particle partitioning schemes have been incorporated into CMAQ. The more complicated schemes apparently provide the best model performance relative to measurement data, but the authors state that the disagreements between measurements and the model are likely mostly attributable to uncertainties in emission estimates. Therefore it is difficult to draw conclusions about which partitioning scheme is best based just on results reported in this paper.

We appreciate the thoughtful review and the recognition of the novelty of the approach in 3D air quality systems such as CMAQ, despite the limited spatiotemporal coverage of the EMEP network.

I have two scientific concerns about the paper. I think both are significant, but possibly #1 arises from unclear presentation in the paper.
1) I understand Table 2 and the text to say that Scenarios 3, 4 and 5 include both the Junge-Pankow adsorption model *and* the Harner-Bidleman Koa absorption model. Is that correct? If so, I think the model is perhaps adding two redundant descriptions of gas-particle partitioning.
It has been a while since I looked at the JP model, but my recollection is that it includes a fitting parameter that was derived from empirical data under the assumption that all partitioning was by adsorption to aerosol surfaces. But, the assumption that adsorption dominates could not be confirmed.

Xiao and Wania (http://www.sciencedirect.com/science/article/pii/S1352231003002139) later showed that vapor pressure and Koa both describe sorption to organic matter equally well. Therefore, I think partitioning schemes that include both vapor pressure and Koa as chemical parameters are likely including redundant information, and could easily be over-fitted.

Reply: We agree, the parameter $c_J$ (high for strong sorption, low for weak sorption) in the JP model may not be totally free from absorptive contributions. Therefore, adding θ of JP and $K_{oa}$ may eventually lead to overpredictions, but these should be very low (negligible), as the absorptive contribution within JP is very low. It is very low, as the fitting was done for all possible vapors, and many of these were inorganic (almost not soluble in octanol). Unfortunately, no $c_J$ for BAP based on measurements has ever been published. However, we found recently in background aerosol of Central Europe (Kosetice 2012-13, n=162 aerosol samples, $c_J$ derived from experimental S/V data; Shahpoury et al., 2016) that $c_J$ for BaP is very low, namely 0.01-5.46 Pa cm (median = 0.44 Pa cm; and also low for all other PAHs, with one exception, anthracene for which $c_J$>17.2 Pa cm is found). This finding is not representative for all types of aerosol in the simulation, but for the European background aerosol. It supports the perception that a possible implicit absorptive contribution to PAHs' $c_J$ must be very low i.e., negligible. Therefore, the combined partitioning according to both the JP and a $K_{oa}$ model is justified (note that the Harner-Bidleman version of the $K_{oa}$ model is present only in scenario 3 − Table was modified to avoid confusion).

New text in the revised version reads (after line 13, page 5): "The empiric JP model, despite assuming adsorption to be the only relevant partitioning process, may eventually not be free from absorptive contributions (to $c_J$), but these should be very low. The fitting in the empiric parameterisation was done for all possible vapours, many of these not soluble in octanol. No $c_J$ for BAP derived from ambient measurements has ever been published. However, we found recently in background aerosol of Central Europe (Kosetice 2012-13, n=162 aerosol samples, $c_J$ derived from experimental S/V data; Shahpoury et al., 2016) that $c_J$ for BAP is very low, namely 0.01-5.46 Pa cm (median 0.44 Pa cm), supporting the perception that a possible implicit absorptive contribution to $c_J$ must be very low."

Shahpoury P., Lammel G., Albinet A., Sofuoglu A., Domanoğlu Y., Sofuoglu C.S., Wagner Z., Zdimal V.: Evaluation of a conceptual model for gas-particle partitioning of polycyclic aromatic hydrocarbons using poly-parameter linear free energy relationships, Environ. Sci. Technol., under review 2016.

2) The authors state that soot-air partition coefficients were calculated as the ratio of soot-water adsorption constants (Ksw) and the Henry's Law constant. Kai Goss has pointed out that applying a thermodynamic triangle for a solid interface in contact with water and the same solid interface in contact with an adjacent gaseous phase is a conceptual mistake that can lead to very large errors (http://pubs.acs.org/doi/abs/10.1021/es0301370).
I actually do not see a good option for the authors to overcome this problem, since the Lohmann & Lammel (2004) and the Dachs & Eisenreich (2000) papers that are the basis for the gas-particle partitioning scheme applied in this paper are both based on thermodynamic triangles that Goss argues (convincingly!) are invalid.

Reply: We thank the reviewer for pointing this out. We agree that the method applied is imperfect, but it is the best available and, hence, commonly accepted in atmospheric modelling (as cited by the referee, Galarneau et al., 2014, Wang et al. Atmos Environ 2013; Wang et al. Environ Sci Pollut Res 2013; Friedman et al Environ Sci Technol 2014; Galarneau et al. Atmos Chem Phys 2014; Sangiorgi et al. Environ Sci Pollut Res 2014; Liu et al.. Chemosphere 2015; Li et al Atmos Environ 2016, and others). The only other option available (suggested by Dachs et al., 2004, and van Noort, 2003) parameterises $K_{soot-air}$ as a function of the BC specific surface area, which is basically unknown and certainly transient while aging (typically changing by at least one order of magnitude within one day following emission from combustion sources). Dachs concludes that prediction of $K_{sa}$ by either method provide reasonably similar results.

New text in the revised version reads (after line 24, page 6): "This method is subject to uncertainties (Goss, 2004), but is accepted and suitable (Dachs et al., 2004; besides others)."

van Noort, P.: A thermodynamics-based estimation model for adsorption of organic compounds by carbonaceous materials in environmental sorbents, Environ. Toxicol. Chem., 22(6), 1179–1188, doi:10.1002/etc.5620220601, 2003.

Dachs, J., Ribes, S., van Drooge, B., Grimalt, J., Eisenreich, S. J. and Gustafsson, Ö.: Response to the comment on "Influence of soot carbon on the soil−air partitioning of polycyclic aromatic hydrocarbons," Environ. Sci. Technol., 38(5), 1624–1625, doi:10.1021/es0307118, 2004.

Goss, K.-U.: Comment on "Influence of soot carbon on the soil−air partitioning of polycyclic aromatic hydrocarbon," Environ. Sci. Technol., 38(5), 1622–1623, doi:10.1021/es0301370, 2004.

**Anonymous Referee #2**

General comments

The manuscript describes the application of community air quality model CMAQ to study atmospheric transport of B(a)P and effects of gas-particle partitioning and degra- dation. Several gas-particle partitioning models were implemented in the CMAQ model extending its capabilities with respect to modeling of PAHs. Though the subject of the study is of importance with regard to existing air quality problems in Europe, description of this study suffers from obscurity and thus requires substantial revisions.

First of all, I would mention not clear enough description of parameters applied in the equations 10 and 11, related to gas-particle partitioning (e.g. KOA, KSA). Secondly, evaluation of the effects of several GPP mechanisms through their incremental testing (incremental testing is defined in section 3.2 and Table 2) is not quite clearly described with regard to sensitivity of CMAQ model output to particular partitioning mechanisms in model simulation scenarios. In some cases abbreviations, defined for GPP models and scenarios and used throughout the analysis, are mixed like e.g. DE model and DE scenario. This leads to problems with understanding what is presented, for example, in the second column of tables 4 and S8 called 'Model'. There are both abbreviations for scenarios and models (for example, JP-W and DE, HB, etc.). Finally, the underestimation of observed B(a)P concentrations in air is explained

by low emissions. At the same time, there is no discussion of potential problems of emission data used, no comparison of total annual emissions with other studies, for example, with similar studies e.g. Aulinger et al. (2007) and others.

Reply: We appreciate the very thorough and detailed review along with the recognition of the potential relevance of the presented study. The first two points of concern have been addressed in a revised manuscript. Regarding the last area of concern, we feel that the manuscript adequately discusses the limitations of emission data used, which have been the major topic of work published by the authors (Bieser et al., 2012).

Specific comments

Page 3, line 12-13: "CMAQ contains modules representing advection, eddy diffusion, in-cloud, and precipitation processes". It would be better to use 'in cloud and below- cloud scavenging with precipitation'.

Reply: Corrected

Page 3, line 26: 'gas phase reactions' instead of 'gas reactions'.

Reply: Corrected

Page 4, line 23: Though the equation 2 follows the publication of Aulinger et al. (2007), it is not clear how the particulate fractions in each mode i, fi, are obtained, because in the cited work (Cooter and Hutzell, 2002) similar equation is written for the sum of partition coefficients, but not for fractions of a compound in particulate phase.

Reply: According to equations 1-4 in Cooter and Hutzell (2002), particulate fractions are defined for each mode i of CMAQ aerosol.

Page 5, equation 7: fOM is used without index i. Does it mean that the fraction of organic matter in aerosol particles is the same in each of three modes?

Reply: Corrected throughout the text

Page 6, line 3: assumptions used here for ratio of activity coefficients and ratio of mean molar weights of organic matter of the particles and octanol (that they are equal unity) need to be discussed with regard to their uncertainties as it was shown in e.g. (doi: 10.5194/acpd-14-21341-2014 ).

Reply: More explanations will be added. The new text reads (after line 7, page 6): "Assuming that octanol imitates organic matter in PM, Harner and Bidleman (1998) suggested that the ratio of $\gamma_{oct}/\gamma_{OM}$ and $M_{oct}/M_{OM}$ can be assumed to be 1. However, it was later suggested that $M_{OM}$ could be much higher,

particularly in secondary organic aerosols containing polymeric structures (Kalberer et al., 2004); a mean value of 500 g mol$^{-1}$ was later suggested by Götz et al. (2007), which results in $M_{oct}/M_{OM}$ of 0.26."

Götz, C. W., Scheringer, M., MacLeod, M., Roth, C. M. and Hungerbühler, K.: Alternative approaches for modeling gas−particle partitioning of semivolatile organic chemicals: model development and comparison, Environ. Sci. Technol., 41(4), 1272–1278, doi:10.1021/es060583y, 2007.
Kalberer, M., Paulsen, D., Sax, M., Steinbacher, M., Dommen, J., Prevot, A. S. H., Fisseha, R., Weingartner, E., Frankevich, V., Zenobi, R. and Baltensperger, U.: Identification of polymers as major components of atmospheric organic aerosols, Science, 303(5664), 1659–1662, doi:10.1126/science.1092185, 2004.

Page 6, line 14-15: Soot-air partition coefficients were calculated as the ratio of soot water adsorption constants KSW and the inverse Henry's Law constant (HC), with KSA values adopted from Bärring et al. (2002). Concerning the way of deriving the KSA it would be important to provide estimates of uncertainties that would be introduced by this assumption. Moreover, it is unclear how this adopting was performed for B(a)P since the publication of Bärring et al. (2002) was focused on experimental determination of the soot−water distribution coefficients for PCDDs, PCDFs, and PBDEs.

Reply: $K_{sa}$ was calculated based on $K_{sw}$ (typo fixed) following the approach described in the footnotes of Table 1 in Lohmann and Lammel (2004). Despite the publication title not mentioning PAHs, this value is also referenced in Table 4 of Bärring et al. (2002). This information will be included in Table 1 for clarity. We agree that the method applied is imperfect, but it is the best available and, hence, commonly accepted in atmospheric modelling (as cited by the referee, Galarneau et al., 2014, Wang et al. Atmos Environ 2013; Wang et al. Environ Sci Pollut Res 2013; Friedman et al Environ Sci Technol 2014; Galarneau et al. Atmos Chem Phys 2014; Sangiorgi et al. Environ Sci Pollut Res 2014; Liu et al.. Chemosphere 2015; Li et al Atmos Environ 2016, and others). The only other option available (suggested by Dachs et al., 2004, and van Noort, 2003) parameterises $K_{soot-air}$ as a function of the BC specific surface area, which is basically unknown and certainly transient while aging (typically changing by at least one order of magnitude within one day following emission from combustion sources). Dachs concludes that prediction of $K_{sa}$ by either method provide reasonably similar results.

New text in the revised version (following a suggestion of the other reviewer) reads (after line 24, page 6):
"This method is subject to uncertainties (Goss, 2004), but is accepted and suitable (Dachs et al., 2004; besides others)."

van Noort, P.: A thermodynamics-based estimation model for adsorption of organic compounds by carbonaceous materials in environmental sorbents, Environ. Toxicol. Chem., 22(6), 1179–1188, doi:10.1002/etc.5620220601, 2003.
Dachs, J., Ribes, S., van Drooge, B., Grimalt, J., Eisenreich, S. J. and Gustafsson, Ö.: Response to the comment on "Influence of soot carbon on the soil−air partitioning of polycyclic aromatic hydrocarbons," Environ. Sci. Technol., 38(5), 1624–1625, doi:10.1021/es0307118, 2004.

Goss, K.-U.: Comment on "Influence of soot carbon on the soil−air partitioning of polycyclic aromatic hydrocarbon," Environ. Sci. Technol., 38(5), 1622–1623, doi:10.1021/es0301370, 2004.

Page 6, line 16: There would be a need to describe more clearly these two parameterizations for $K_{oa}$ (e.g. to give equations, to show difference).

Reply: The two $K_{oa}$ parameterisations are described in the cited publications. Previous modelling studies of BaP followed either Beyer et al. (Aulinger, 2007) or Odabasi et al. (Galarneau, 2014). The relevant text was amended to explicitly state that Odabasi et al. determine $K_{oa}$ as a function of temperature.

Page 10, line 6: It would be better to provide the difference (relative or absolute) between the maps in Figures 2a and 2b instead of direct comparison.

Reply: The goal of Figure 2 is to provide an illustration of BaP distributions and gradients across Europe. For better clarity we selected logarithmic scale. The difference is in fact provided in Figure 3 that provides the absolute difference comparison plot, discussed in the following paragraph.

Page 21, Table 1: it is not shown from where the values for $K_{soot-water}$, $K_{soot-air}$, and OH reaction rate were taken.

Reply: Corrected

Page 22, table 2: Table 2 does not correspond to its description in section 3.2 for 4th and 5th scenarios.

Reply: Corrected

Page 27, figure 3: Please, correct the abbreviations JB-W and DL, as they were not defined earlier. The same for figure S5. Concerning the maps, it would better to show the difference between them in relative units.

Reply: Corrected the abbreviations uniformly throughout the text. However, no visual benefit when moving from absolute to relative units was noticeable in the figures, therefore omitted.

Table 4 and Table S8 do not present mean modeled and observed B(a)P concentrations. In the column 'Model' both abbreviations for scenarios and for GPP models are used (for example, JP-W and DE, HB, etc.). Please, provide equations or references for IOA etc.

Reply: Corrected the abbreviations throughout the text and according to previous comment. The metrics of performance have been calculated based on the openair R package (Carslaw and Ropkins, 2012) as noted in section 3.2 (equations/references can be found therein).

[revised manuscript text omitted]